# Compositional adaptability in NPM1-SURF6 scaffolding networks enabled by dynamic switching of phase separation mechanisms

Mylene C. Ferrolino[1], Diana M. Mitrea[1], J. Robert Michael[2] & Richard W. Kriwacki [1,3]

The nucleolus, the site for ribosome biogenesis contains hundreds of proteins and several types of RNA. The functions of many non-ribosomal nucleolar proteins are poorly understood, including Surfeit locus protein 6 (SURF6), an essential disordered protein with roles in ribosome biogenesis and cell proliferation. SURF6 co-localizes with Nucleophosmin (NPM1), a highly abundant protein that mediates the liquid-like features of the granular component region of the nucleolus through phase separation. Here, we show that electrostatically-driven interactions between disordered regions of NPM1 and SURF6 drive liquid-liquid phase separation. We demonstrate that co-existing heterotypic (NPM1-SURF6) and homotypic (NPM1-NPM1) scaffolding interactions within NPM1-SURF6 liquid-phase droplets dynamically and seamlessly interconvert in response to variations in molecular crowding and protein concentrations. We propose a mechanism wherein NPM1-dependent nucleolar scaffolds are modulated by non-ribosomal proteins through active rearrangements of interaction networks that can possibly contribute to the directionality of ribosomal biogenesis within the liquid-like nucleolus.

[1] Department of Structural Biology, St. Jude Children's Research Hospital, Memphis, TN, USA. [2] Department of Computational Biology, St. Jude Children's Research Hospital, Memphis, TN, USA. [3] Department of Microbiology, Immunology and Biochemistry, University of Tennessee Health Sciences Center, Memphis, TN, USA. Correspondence and requests for materials should be addressed to R.W.K. (email: richard.kriwacki@stjude.org)

The nucleolus, a membrane-less organelle with active, liquid-like physical properties[1], is the site for ribosome biogenesis as well as a signaling hub for cell cycle regulation and cellular stress responses[1–5]. Three immiscible phases, the fibrillar center (FC), dense fibrillar component (DFC), and granular component (GC), compartmentalize the nucleolus[2,3,6,7]. Originating in the FC, where ribosomal DNA, the genes for ribosomal RNA (rRNA) are clustered and transcribed, pre-rRNA is post-transcriptionally modified and processed within the DFC, followed by assembly with ribosomal proteins within the GC[7]. The nucleolus contains over 700 ribosomal, non-ribosomal, and accessory proteins[6,8], and several types of RNA[7]. While ribosomal proteins partition into the nucleolus for assembly into ribosomal subunits, and the proteins fibrillarin (FIB) and nucleophosmin (NPM1) contribute to the organization of the DFC and GC compartments through phase separation[9–11], the roles of many other non-ribosomal nucleolar proteins are poorly understood. For example, Surfeit locus protein 6 (SURF6), encoded by the *Surfeit* locus, is an essential non-ribosomal nucleolar protein[12,13] that promotes ribosome biogenesis and cell proliferation when overexpressed[14], but the underlying functional mechanism is unknown.

Human SURF6, comprised of 361 amino acids, is predicted to be intrinsically disordered and exhibits multiple arginine-rich short linear motifs (termed R-motifs) within its primary structure (Fig. 1a). SURF6 directly interacts and co-localizes with NPM1 in the GC[10,14,15]. NPM1 (Fig. 1b) is a pentameric protein with three distinct domains: (1) an N-terminal oligomerization domain (OD; residues 1–119), (2) a central, polyampholytic intrinsically disordered region (IDR; residues 120–240) with two acidic tracts (A2 and A3) and two basic tracts (B1 and B2), and (3) a C-terminal nucleic acid binding domain (CTD; residues 241–294). Multivalent R-motifs within a disordered N-terminal fragment of SURF6 (S6N; residues 1–182)[10] interact with the two acidic tracts within the IDR of NPM1 (Fig. 1); above the saturation concentration, in the micromolar range, these interactions cause heterotypic liquid–liquid phase separation (LLPS)[9,10]. Interestingly, two competing mechanisms are active within NPM1-S6N liquid-like droplets: heterotypic LLPS of NPM1 with S6N (forming NPM1-S6N molecular networks) and homotypic LLPS of NPM1 (forming NPM1-NPM1 molecular networks). S6N, through its R-motifs, binds the acidic tracts of NPM1. In this scenario, the basic tracts, especially B2 of NPM1, compete with R-motifs of S6N for binding to NPM1's acidic tracts, especially A3, to promote homotypic LLPS[10]. We previously hypothesized that NPM1's ability to undergo multiple types of LLPS with different classes of nucleolar components (e.g., R-motif-containing ribosomal and non-ribosomal proteins, rRNA, and itself) plays a buffering role to maintain the liquid-like structural scaffold of the nucleolar GC. This buffering capacity may compensate for variations in the constellation of NPM1 partners present in the nucleolus as pre-ribosomal particles vectorially assemble from the FC at the center towards the GC at the periphery[10]. Here, we show that the compositional and physical properties of NPM1-S6N droplets are modulated by competition between the NPM1-S6N heterotypic and NPM1-NPM1 homotypic scaffolding mechanisms and that interplay between these mechanisms allows dynamic and seamless adaptation to changes in the concentrations of NPM1's partners and the extent of molecular crowding.

## Results

**Molecular crowding promotes NPM1-S6N phase separation.** The cell interior is highly crowded, with total macromolecular concentrations between 100 to 300 mg·mL$^{-1}$[16] and viscosities between ~1 and 50 mPa·s[17]. Molecular crowding increases viscosity of solutions and consequentially reduces molecular diffusion, and is known to induce polymer chain compaction and stabilize molecular interactions[16]. Previously, we reported that NPM1 undergoes homotypic LLPS only under crowded conditions, indicating that molecular crowding promotes NPM1-NPM1 interactions and reduces the saturation concentration, above which phase separation occurs[10]. Based on this observation, we hypothesized that the saturation concentration for heterotypic LLPS of NPM1 with S6N would be reduced in a crowding agent concentration-dependent manner. To test this hypothesis, we monitored phase separation by measuring solution turbidity for binary mixtures of NPM1 and S6N in the presence of variable concentrations of organic polymers. As predicted, the NPM1 and S6N saturation concentrations shifted to lower values in the presence of 5 and 15% polyethylene glycol-8000 (PEG), with respect to non-crowded buffer conditions (Fig. 2a). NPM1 underwent homotypic phase separation at the crowding agent concentrations tested, but S6N did not (Fig. 2a). Saturation concentrations for LLPS were similarly reduced in the presence of other polymeric crowding agents, including Ficoll-70 and Dextran-10000 (Supplementary Fig. 1).

Next, we quantified the protein concentrations in the dense phases using confocal microscopy imaging of LLPS droplets prepared with Alexa Fluor 488-labeled NPM1 (NPM1-A488) and Alexa Fluor 647-labeled S6N (S6N-A647) (see Methods and Supplementary Figs. 2 and 3). The concentration of proteins within the droplets was 100–200 mg·mL$^{-1}$, which was within the concentration range of the crowding agents used to induce homotypic NPM1 LLPS (50 or 150 mg·mL$^{-1}$; Fig. 2a). With increasing crowding agent concentrations (Fig. 2b), the NPM1:S6N molar ratio within droplets remained constant at approximately 10:1 (Fig. 2c), corresponding to one S6N molecule interacting with two NPM1 pentamers. We recently showed that, under uncrowded buffer conditions, the ability of NPM1 to undergo homotypic, in addition to heterotypic phase separation is associated with its enrichment within droplets comprised of NPM1 and S6N[10]. Our observation that the NPM1-S6N molar ratio does not change with the crowding agent concentration suggests that that homotypic and heterotypic LLPS mechanisms are affected similarly by molecular crowding.

**Crowding stabilizes inter-molecular contacts within droplets.** As previously demonstrated[10], uncrowded heterotypic droplets with NPM1 and S6N exhibit liquid-like properties and in fluorescence recovery after photobleaching (FRAP) assays NPM1 fluorescence recovers almost fully within 2 min (Fig. 2d).

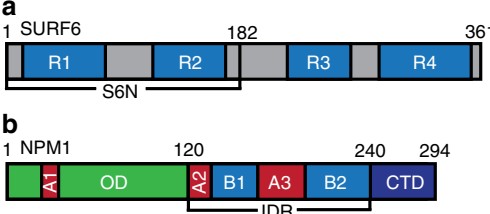

**Fig. 1** Structural features of NPM1 and S6N. **a** SURF6 is a disordered protein which consists of four arginine-rich motifs (R-motifs), R1–R4 (blue). An N-terminal construct (S6N; residues 1–182), which includes two of the R-motifs, R1 and R2, was used in this study. **b** NPM1 is comprised of an oligomerization domain OD (green), a long intrinsically disordered region (IDR) spanning residues 120–240, and a C-terminal nucleotide binding domain (CTD). Acidic and basic tracts in the IDR, A2 and A3 (red), and B1 and B2 (blue), respectively, and the CTD (dark blue), are involved in electrostatic interactions

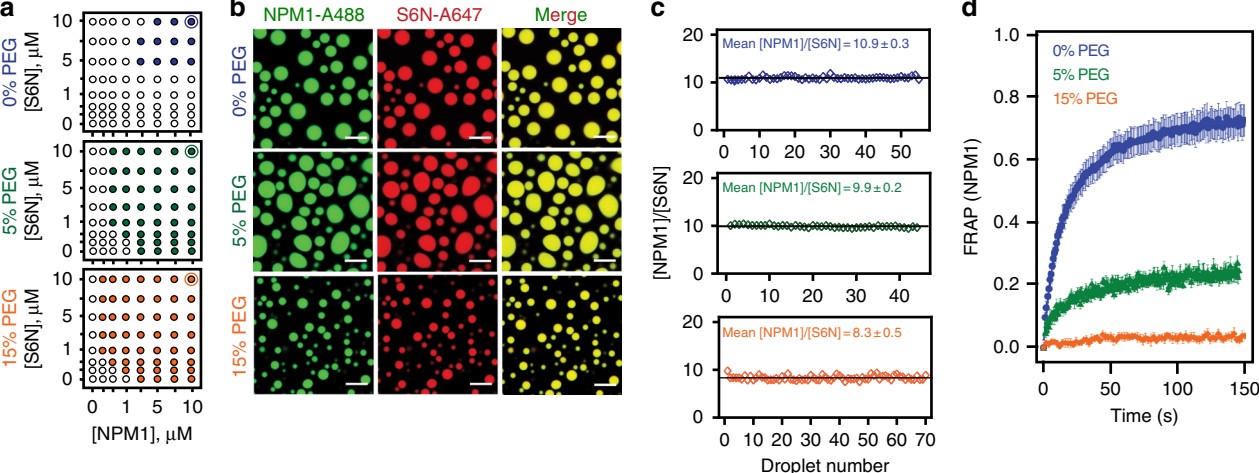

**Fig. 2** Effects of macromolecular crowding on homotypic NPM1 and heterotypic NPM1-S6N LLPS. **a** Phase diagrams for LLPS of mixtures of NPM1 and S6N in the presence of different concentrations of PEG (0%, 5%, and 15% PEG, as indicated) determined by turbidity assays. Phase separation was not observed for protein concentrations represented by open circles ($OD_{340}$ <0.1) and was observed for those represented by solid colored circles ($OD_{340}$ ≥0.1). The solutions also contained 10 mM Tris, 150 mM NaCl, 2 mM DTT, pH 7.5 buffer. **b** Confocal microscopy images of NPM1-A488 (green) and S6N-A647 (red) droplets in the presence of different PEG concentrations (0%, top; 5%, middle; and 15%, bottom); scale bar = 10 μm. NPM1 and S6N concentrations were 10 μM each represented in the phase diagrams in **a** by thin blue, green, and orange circles around the solid circles of similar color. **c** Ratios of concentrations of NPM1 and S6N within individual NPM1-S6N droplets. NPM1-SURF6 molar concentration ratios ([NPM1]/[S6N]) for droplets prepared in the presence of different concentrations of PEG (0%, top panel; 5%, middle panel; or 15%, bottom panel; $n \geq 45$). The black line indicates the average [NPM1]/[S6N] ratio for droplets at the specified crowding agent concentrations. In these experiments, concentrated PEG was added to pre-mixed solutions (in 10 mM Tris, 150 mM NaCl, 2 mM DTT, pH 7.5 buffer) of NPM1 and S6N to give the noted final concentrations. **d** FRAP curves for NPM1 for NPM1-S6N droplets prepared in the presence of different PEG concentrations (0%, blue; 5%, green; or 15%, orange) after incubation for 4 h; ROI = 1 μm circular area in the center of the droplet. Values represent mean ± s.d

**Table 1 FRAP fraction recovery and half recovery times of NPM1 in NPM1-S6N droplets with and without crowding agent**

| Buffer | $\eta$ buffer (mPa·s) | Mf | $t_{1/2}$ (s) | $D_{app}$ (μm$^2$·s$^{-1}$) |
|---|---|---|---|---|
| 0% PEG | 1.0 ± 0.001 | 0.79 ± 0.001 | 14.3 ± 0.2 | $2.9 \times 10^{-3} \pm 4.0 \times 10^{-5}$ |
| 5% PEG | 2.4 ± 0.001 | 0.27 ± 0.003 | 22.5 ± 1.7 | $2.1 \times 10^{-3} \pm 1.3 \times 0^{-4}$ |
| 15% PEG | 10.7 ± 0.02 | ~0.05 | ND | ND |

All solutions contain 10 mM Tris, 150 mM NaCl, 2 mM DTT, pH 7.5
$\eta$ viscosity, Mf fraction of mobile NPM1 from FRAP, $t_{1/2}$ half-time of fluorescence recovery,
$D_{app}$ apparent diffusion coefficient, mean ± s.d.; $n \geq 9$ FRAP measurements; $n = 4$ for buffer viscosity determination; ND cannot be determined from fitted FRAP curves

However, the introduction of molecular crowding dramatically decreased NPM1 mobility, with its mobile fraction dropping from ~80% in buffer without crowding agent to ~30% and ~5% in buffer with 5 and 15% PEG, respectively (Figure 2d and Table 1). While an overall decrease in molecular diffusion is expected due to an increase in the viscosity of the PEG-containing solutions (≤10-fold compared to buffer alone; Table 1), this effect alone does not explain the decrease in mobility of the NPM1 mobile fraction. For example, an increase of the NPM1-S6N droplet viscosity by 100-fold—much greater than that associated with 15% PEG (see Table 1)—would still allow for 15% NPM1 fluorescence recovery in the same experimental time frame (Supplementary Fig. 4); consequently, we propose that the reduced NPM1 mobility associated with crowding is due to increased inter-molecular interactions, which effectively mediate non-covalent cross-linking within the scaffold of the NPM1-S6N droplets. We hypothesize that PEG-induced molecular crowding promotes enhanced homotypic, inter-NPM1 pentamer interactions, resulting in increasingly interconnected NPM1-NPM1 networks. We next characterized homotypic phase separation by NPM1 in the presence of different concentrations of PEG to deconvolute the contributions of the heterotypic and homotypic

mechanisms to the properties of the NPM1-S6N droplets discussed above.

**Networks within NPM1 droplets stabilize over time.** We hypothesized that increased molecular crowding at higher PEG concentrations would raise the effective concentration of NPM1 within homotypic droplets due to volume exclusion effects. As a consequence, crowding would promote NPM1-NPM1 interactions through increased branching of NPM1 networks, and reduce NPM1 dynamics. In fact, the concentration of NPM1 within homotypic droplets positively correlated with the PEG concentration (Fig. 3a; Supplementary Table 1). Consistent with the increased concentration of molecules incorporated in the dense-phase, light-phase concentrations decreased (Fig. 3a; Supplementary Table 1). Two distinct mechanisms could be envisioned that result in enhanced partitioning of NPM1 molecules within the dense phase. For one potential mechanism, PEG is an inert crowding agent, which causes chain compaction and stabilizes the electrostatic interactions between A- and B-tracts of NPM1 through volume exclusion effects, thereby driving the accumulation of NPM1 within the dense phase. For a second

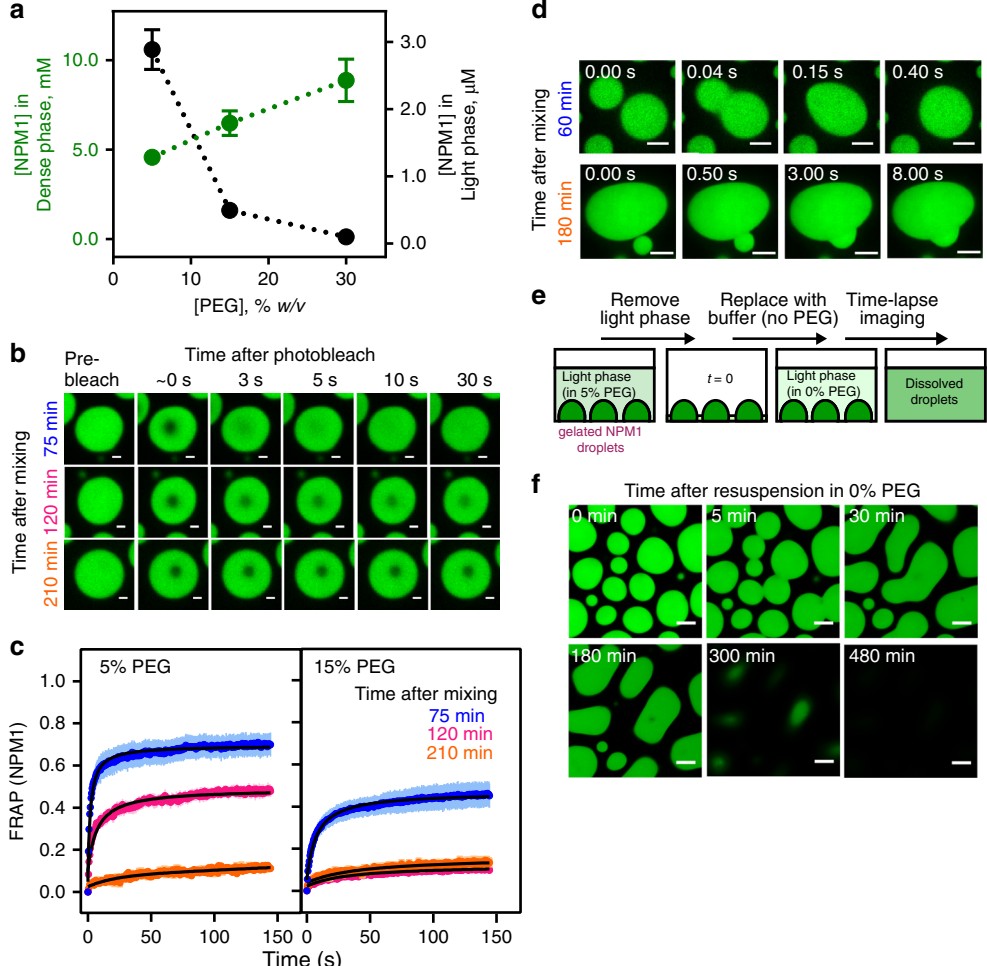

**Fig. 3** Homotypic NPM1 droplets undergo reversible aging. **a** NPM1 concentrations measured within the dense (green, $n \geq 343$ droplets) and light (black, $n = 6$) phases prepared in the presence of 5%, 15% and 30% PEG (see Methods). Values represent mean ± s.d. **b** Confocal microscopy images of NPM1-A488 droplets photobleached at $t = 75$ min (top row), 120 min (middle row), and 210 min (bottom row) after mixing 20 μM NPM1 with 5% PEG in 10 mM Tris, 150 mM NaCl, 2 mM DTT, pH 7.5 buffer; scale bar = 1 μm. **c** FRAP recovery curves of NPM1-A488 in 20 μM NPM1 droplets at $t = 75$ min (blue), 120 min (magenta), and 210 min (orange) after droplet formation in the presence of 5% PEG (left panel) and 15% PEG (right panel). Values represent mean ± s. d. for $n \geq 8$ droplets. Under both crowding agent conditions, the fraction of NPM1 that recovered decreased at the later time points after mixing; ROI = 1 μm circular area in the center of the droplet. **d** Time-lapse fluorescence microscopy images of fusion between NPM1-A488 droplets formed 60 min (top row) or 180 min (bottom row) after mixing in the presence of 5% PEG. Droplets fused rapidly after incubation for 60 min, but fusion was very slow after incubation for 180 min. **e** An illustration of the experimental scheme used to monitor dissolution of aged NPM1 droplets formed in 5% PEG upon removal of the crowding agent. Light phase (90% of the solution) was gently removed and replaced with the same volume of buffer lacking PEG. Droplet dissolution was monitored over time using confocal fluorescence microscopy imaging. **f** Time-lapse imaging of aged droplets dissolving after removal of the crowding agent

potential mechanism, PEG directly interacts with NPM1, and drives NPM1 accumulation within the dense phase by forming a heterotypic PEG-NPM1 scaffold. One fundamental feature of the NPM1 droplets that can discriminate between these two mechanisms is PEG partitioning. An inert crowding agent would be uniformly distributed between the light and dense phase (or be excluded from the dense phase if its dimensions are greater than the effective mesh size of the scaffold[18]), while a scaffolding crowding agent would be enriched within the dense phase. Using TAMRA-labeled PEG-10K (PEG-TAMRA) as a probe, we measured partition coefficients of 0.7 ± 0.1, 0.9 ± 0.1, and 0.9 ± 0.1 within homotypic NPM1 droplets formed in the presence of 5%, 15%, and 30% PEG, respectively (Supplementary Fig. 5; Supplementary Table 2; see also Supplementary Methods), indicating that PEG molecules do not favorably bind to NPM1 to form NPM1-PEG scaffolding interactions. To further demonstrate that PEG is not part of the scaffold, we tested the PEG chain length

dependence of LLPS. We compared NPM1 LLPS in solutions containing PEGs of different molecular weights (1, 4, 8, and 20 kDa). We found that PEG-induced LLPS occurs at a minimum molecular weight of ~4 kDa, and larger PEGs do not further enhance phase separation (Supplementary Fig. 6). Interestingly, this range of PEG sizes (e.g., >4 kDa) correlated with the previously reported long-chain regime of crowders that can induce compaction of IDPs[19], suggesting that NPM1-IDR chain compaction may contribute to PEG-dependent phase separation. We conclude that PEG-dependent volume exclusion promotes NPM1-NPM1 interactions and homotypic LLPS, and may contribute to the immobilization of NPM1 within NPM1-S6N heterotypic droplets (Fig. 2D). Notably, we also observed that the extent of recovery of NPM1 fluorescence after photobleaching was time dependent. For example, under moderate crowding with 5% PEG, within 75 min after formation of homotypic droplets, ~70% recovery of NPM1 fluorescence was observed (Fig. 3b, c).

However, the recovery of NPM1 fluorescence dropped to ~49% and ~14% after incubating for 120 and 210 min, respectively (Fig. 3b, c). The observed decrease in mobile NPM1 was more pronounced in the presence of 15% PEG, where at 75 min the mobile fraction was only about 47% and further dropped to ~12% within 120 min (Fig. 3b, c). In addition, we observed that the fluorescence intensity of NPM1 within these droplets exhibited only a modest increase over time (Supplementary Fig. 7), indicating that the decrease in molecular mobility was not associated with time-dependent accumulation of NPM1 within the dense phase. Alternatively, we propose that the extent of NPM1-NPM1 interactions increases over time, leading to reduced NPM1 mobility. Formation of more extensive NPM1-NPM1 interactions over time was supported by the results of cross-linking experiments using the amine-reactive cross-linking agent, DSP (Lomant's reagent). A larger percentage of NPM1 was found in covalently cross-linked complexes, upon a brief treatment of droplets with DSP, after incubation for 24 h vs. shorter incubation times of 5 or 60 min (Supplementary Fig. 8). In addition to reduced NPM1 mobility, homotypic NPM1 droplets incubated for a longer time under crowded conditions, exhibited incomplete fusion events (60 min vs. 180 min) (Fig. 3d), suggesting that they underwent aging, as observed for other phase-separated systems[20–23]. In contrast to several other phase separation-prone proteins associated with neurodegenerative diseases (e.g., FUS[22] and hnRNPA1[20]), the aged homotypic NPM1 droplets did not further transition to fibrillar structures after overnight incubation in the presence of 5% PEG. Instead, when the aged droplets were re-suspended in buffer lacking crowding agent, they fully dissolved within 8 h (Fig. 3e, f). In order to verify that the dissolution was not due to perturbation of the equilibrium between the protein concentrations in the dense and light phases, we performed the same experiment using buffer containing 5% PEG (Supplementary Fig. 9). Under these conditions, droplet dissolution was not observed.

We previously showed that crowding agent-induced homotypic LLPS of NPM1 is driven by electrostatic interactions between NPM1 pentamers, mediated by the A3- and B2-tracts within NPM1's IDR[10]. We propose that a minimal network of inter-pentamer interactions drives initial LLPS and that additional interactions form over time, progressively immobilizing NPM1 pentamers and rigidifying the dense phase. The high valency of A- and B-tracts within NPM1 pentamers enables this behavior and we propose that the reversibility of droplet aging is due to the transient nature of the individual electrostatic A- and B-tract interactions that underlie phase separation. These interactions contrast with the ones driving phase separation of FUS and hnRNPA1, which involve motifs containing hydrophobic residues that form β-kinked secondary structure elements;[22,24–26] these are more stable than NPM1's electrostatic interactions, resulting in irreversible droplet aging and fibril formation for disease-associated mutant forms of these proteins[20,22,24]. Notably, despite displaying two basic tracts (B1 and B2; residues 133–160 and 188–240, respectively), the NPM1-IDR is enriched in lysine residues, with few arginine residues (the IDR contains 19 lysine and 3 arginine residues). The content of hydrophobic residues is also low, as the IDR contains two isoleucine, three leucine, two phenylalanine, and four valine residues. Notably, arginine and aromatic residues are often associated with phase separation-prone protein regions[27,28]. These unconventional sequence features may underlie the transience of inter-NPM1 pentamer interactions and the reversibility of homotypic droplet aging. These observations regarding the lability of NPM1-NPM1 interactions within homotypic droplets suggest that NPM1 is able to respond to changing conditions within the nucleolus, for example, due to an influx of R-motif-rich proteins such as SURF6, through scaffold rearrangement. We tested this hypothesis through the experiments discussed below.

**SURF6 alters the architecture and dynamics of NPM1 droplets.** In order to determine how the homotypic NPM1 phase separation scaffold responds to incorporation of R-motif-containing proteins, we prepared droplets with varied ratios of NPM1 (maintained at 5 μM, with 10% labeled with Alexa Fluor 488) and S6N (with 10% labeled with Alexa Fluor 647) in the presence of 5% PEG (Fig. 4a). As the starting concentration of S6N increased, so did its concentration within droplets (Fig. 4b). This suggests that NPM1 responds to a broad range of concentrations of R-motif-containing proteins to form phase-separated scaffolds with different blends of homotypic and heterotypic interactions. FRAP analysis of the droplets formed with different S6N:NPM1 ratios revealed that a large portion of NPM1 molecules are immobile at all ratios. However, NPM1 became more mobile inside droplets with higher S6N:NPM1 ratios, as indicated by the increased mobile fraction (32% for homotypic vs. 70% for a S6N:NPM1 of 4:1) (Fig. 4c, Supplementary Table 3). These results suggest that the diffusing NPM1 molecules are less dynamic within the homotypic vs. the heterotypic scaffold. Replacement of some of the fivefold branched NPM1 polymers ($R_g = 55$ Å[10]) with linear S6N polymers ($R_g = 34$ Å, based on SAXS measurements) is likely responsible for the change in NPM1 dynamics within the liquid-like molecular network when S6N is introduced. Similarly, the mobile fraction of S6N also increased from 0.64 to 0.97 with the increase in S6N:NPM1 ratios (Fig. 4a, Supplementary Table 3), indicating that, while S6N remained dynamic irrespective of the S6N content, the nature of the scaffold experienced changes. We also note that similar to purely homotypic droplets, heterotypic droplets also undergo aging to a lesser extent (Supplementary Figure 10). Thus, our results indicate that within NPM1-S6N droplets, heterotypic and homotypic interactions occur simultaneously, leading to mixed scaffolds that depend upon the initial S6N:NPM1 ratio.

Interestingly, the order of reagent addition affected droplet composition. For Fig. 4a, the droplets were prepared by adding crowding agent to pre-mixed NPM1 and S6N, which yielded droplets of uniform composition at each of the different S6N:NPM1 ratios. In contrast, addition of NPM1 to S6N dissolved in buffer with 15% PEG at the 1:1 S6N:NPM1 ratio yielded droplets of heterogeneous composition, ranging from the equilibrium blend of ~10:1 NPM1:S6N to NPM1-enriched droplets with NPM1:S6N mole ratios up to ~15:1 (Supplementary Fig. 11). These results suggest that, as the added NPM1 molecules mix with PEG-dissolved S6N, they stochastically either undergo primarily homotypic or heterotypic LLPS, giving rise to droplets with the observed heterogeneous NPM1-S6N blends. These results demonstrate that the initial, stochastic LLPS processes occur fast, kinetically trapping the droplets with non-equilibrium blends.

In summary, our data show that NPM1 forms heterotypic droplets when mixed with S6N over a wide range of concentrations giving rise to molecular networks derived from a blend of homotypic (NPM1-NPM1) and heterotypic (NPM1-S6N) interactions. Next, we probed the temporal responsiveness of the NPM1 networks to changes in the concentration of S6N, as discussed below.

**The NPM1 scaffold dynamically responds to influx of SURF6.** Based on our findings that the composition of heterotypic NPM1-S6N droplets was influenced by the starting concentration of the two proteins, we further asked whether pre-formed homotypic

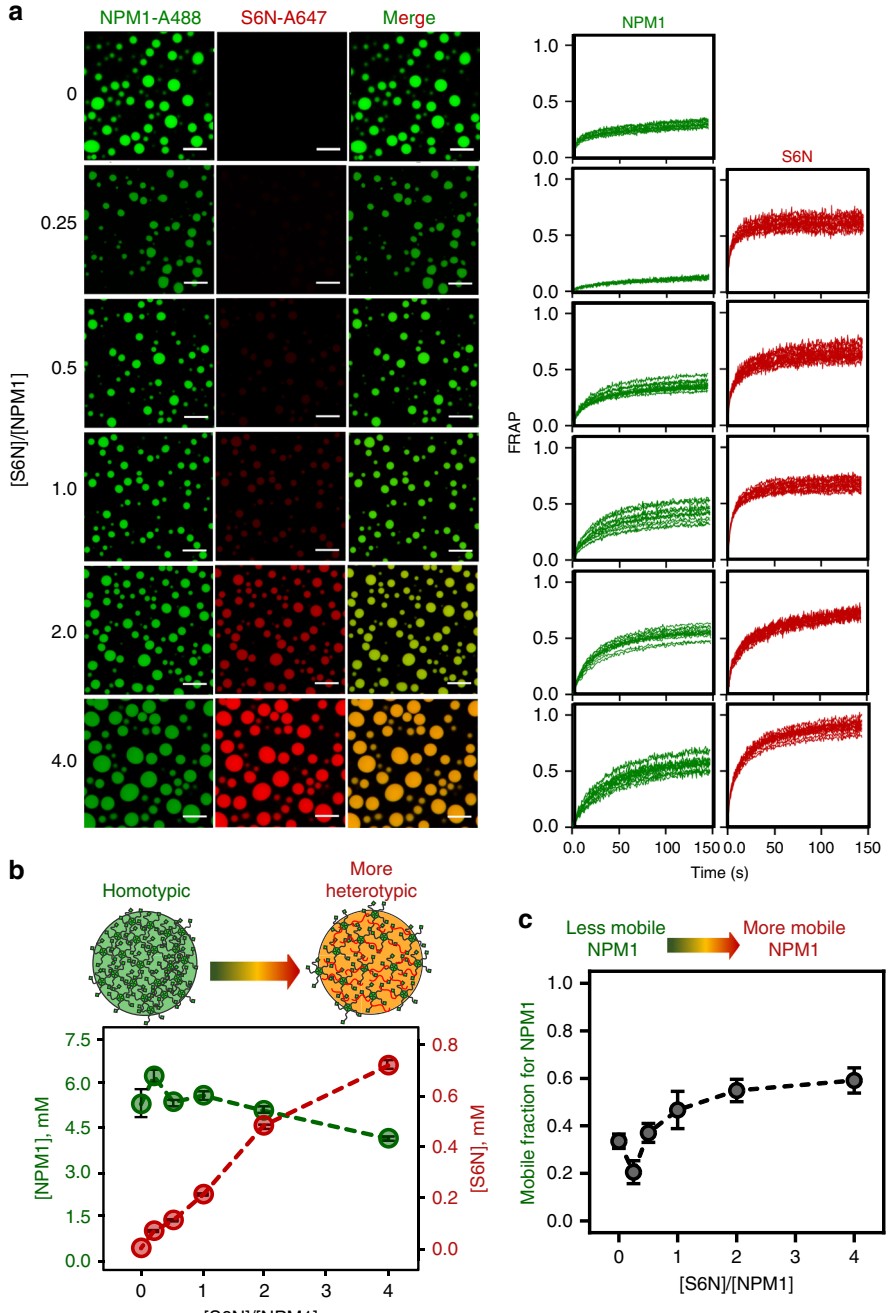

**Fig. 4** S6N tunes the composition and material properties of NPM1 droplet scaffolds. **a** Fluorescence confocal microscopy images (left three columns) of droplets formed from different ratios of NPM1-A488 (left, green) and S6N-A647 (middle, red) in the presence of 5% PEG; the colors of the left and middle columns are merged in the right columns. The concentration of NPM1 (5 μM) was constant and that of S6N was varied, as given by the [S6N]/[NPM1] values indicated on the left. FRAP recovery curves (right two columns) for NPM1-A488 (left, green) and S6N-A647 (right, red), respectively, for individual droplets at the corresponding [S6N]/[NPM1] values, where NPM1 (10 μM) was constant; ROI = 1 μm circular area in the center of the droplet. **b** Concentrations of NPM1 (green) and S6N (red) within droplets prepared at the indicated initial [S6N]/[NPM1] values in the presence of 5% PEG. The starting concentration of NPM1 (5 μM) was constant. Protein concentrations inside droplets were determined using standard curves for both Alexa Fluor 488 and Alexa Fluor 647 in 5% PEG-containing buffer. Values represent mean ± s.d. for $n \geq 20$ droplets. **c** Fraction of mobile NPM1-A488 molecules within droplets derived from FRAP prepared at the different [S6N]/[NPM1] values illustrated in **b**. Values represent mean ± SD for $n \geq 10$ droplets

NPM1 or blended heterotypic NPM1-S6N droplets would dynamically respond to influx of additional S6N molecules in the light phase of the demixed solutions. In order to eliminate variability associated with time-dependent behavior (e.g., droplet aging), we allowed homotypic NPM1 and heterotypic NPM1-S6N droplets to reach equilibrium by incubating them overnight at

room temperature. First, we tested the time dependence of droplet composition upon addition of a sub-stoichiometric amount of S6N (0.5 molar equivalents with respect to NPM1 monomer) to the light phase of aged homotypic NPM1 droplets (prepared with 5% PEG). Over a period of 15 h, S6N slowly accumulated within the homotypic scaffold with little change in the

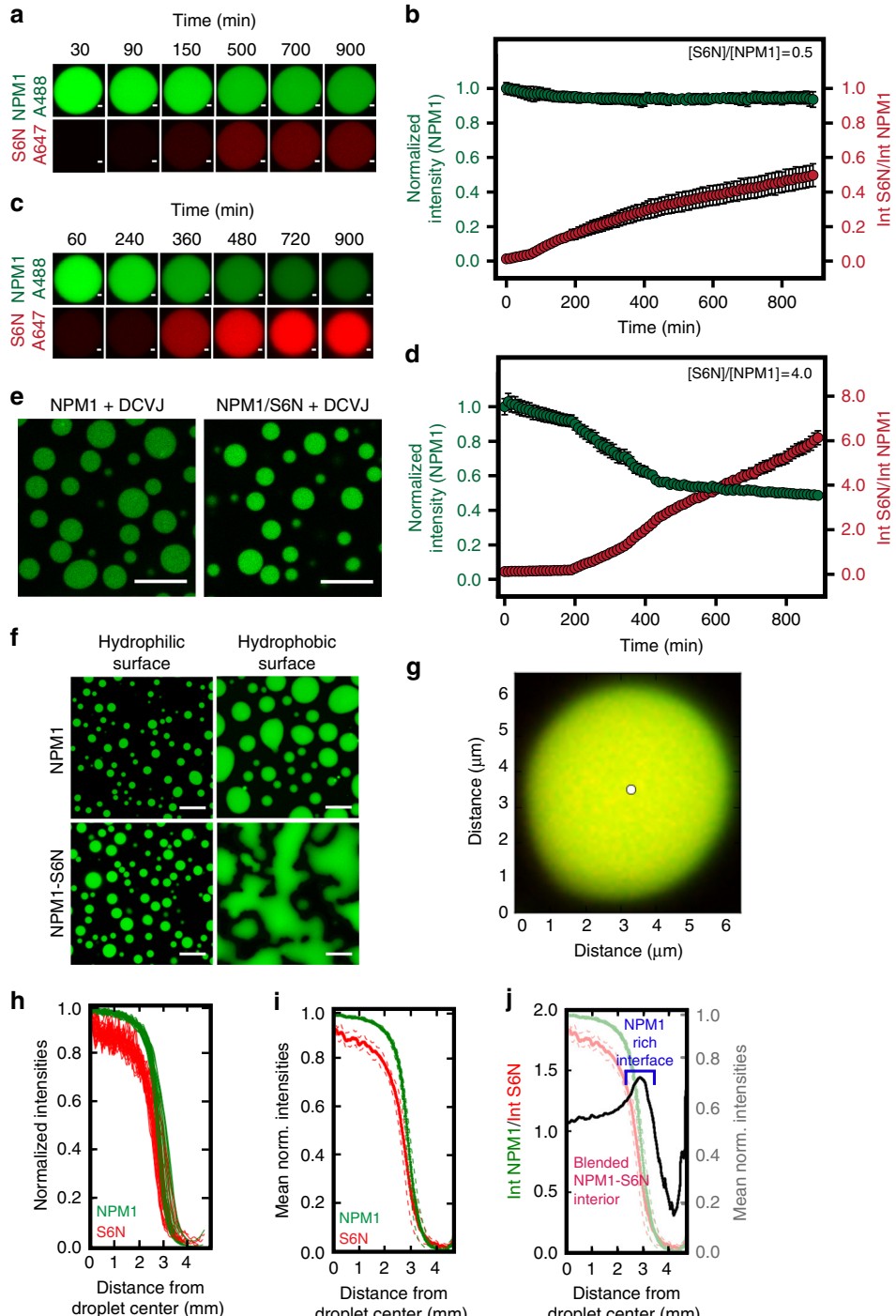

concentration of NPM1 (Fig. 5a, b). Based on these results, we conclude that, at this sub-stoichiometric molar ratio (0.5:1.0 S6N: NPM1), S6N is recruited to the aged NPM1-NPM1 droplets through interactions with available A-tract binding sites within the homotypic scaffold.

We next tested whether the addition of excess S6N would partially replace NPM1 in the scaffold. To do this, we initially equilibrated aged homotypic NPM1 droplets with a sub-stoichiometric amount of S6N (0.5:1.0 S6N:NPM1). This step, together with overnight incubation, was necessary to prevent the formation of new droplets through phase separation of NPM1 in the light phase upon addition of excess S6N. To these pre-equilibrated homotypic NPM1 droplets, which incorporated

client S6N, we added an extra 3.5 mole equivalents of S6N and monitored their composition using time-lapse imaging. Upon addition of this extra amount, S6N was slowly incorporated within the scaffold, as demonstrated by an increase in the S6N-A647:NPM1-A488 fluorescence intensity ratio over time (Fig. 5c, d). Furthermore, S6N competed for pre-existing NPM1-NPM1 interactions, resulting in a decrease in NPM1-Alexa Fluor 488 intensity within droplets (Fig. 5d). This S6N-dependent, partial expulsion of NPM1 from the NPM1-S6N droplets concomitantly triggered formation of de novo, heterotypic droplets with S6N molecules in the light phase (Supplementary Fig. 12). NPM1 was displaced from the initially predominantly homotypic droplets because S6N has higher affinity for the IDR of NPM1 ($K_D = 350$

**Fig. 5** Dynamic scaffolds within homotypic NPM1 and blended heterotypic NPM1-S6N droplets. **a** Time-lapse fluorescence images of an aged NPM1-A488 droplet in 5% PEG after the addition of S6N-A647 at [S6N]/[NPM1] = 0.5; scale bar = 1 μm. NPM1-A488 (top row, green) and S6N-A647 (bottom row, red) are shown. **b** Quantification of the normalized NPM1-A488 fluorescence intensity (green trace, left axis) and S6N-A647/NPM1 fluorescence intensities (red trace, right axis) over time within droplets illustrated in **a**. Values represent mean ± s.d. for $n \geq 10$ droplets. **c** Time-lapse fluorescence images of aged heterotypic NPM1-A488-S6N-A647 droplet (pre-incubated with S6N to [S6N]/[NPM1] = 0.5) in 5% PEG after the addition of S6N-A647 at a final [S6N]/[NPM1] of 4.0; scale bar = 1μm. Droplets are illustrated as in **a**. **d** Quantification of the normalized NPM1-A488 fluorescence intensity (green trace, left axis) and S6N-A647/NPM1-A488 fluorescence intensities (red trace, right axis) over time within droplets illustrated in **c**. Values represent mean ± s.d. for $n \geq 10$ droplets. **e** Fluorescence images of homotypic NPM1 and heterotypic NPM1-S6N droplets ([S6N]/[NPM1] = 4.0) treated with 10 μM DCVJ. **f** Fluorescence images of homotypic NPM1 (top panels) and heterotypic [S6N]/[NPM1] = 4.0, NPM1-S6N droplets (bottom panels) deposited on hydrophilic (left panels) and hydrophobic slide surfaces (right panels); scale bar = 10 μm. **g** Composite of fluorescence confocal microscopy images of NPM1-A488 (green) and S6N-A647 (red) of an NPM1-S6N droplet at [S6N]/[NPM1] = 1.0 in 5% PEG showing an NPM1-rich outer layer. The white circle marks the droplet center determined by Sauron algorithm. **h** Normalized radial intensities for NPM1-A488 and S6N-A647 quantified around one NPM1-S6N droplet (from **g**) plotted vs. the distance from the center of the droplet determined using the Sauron algorithm. **i** Mean normalized radial intensities (solid lines) for NPM1-A488 and S6N-A647 for NPM1-S6N droplet plotted vs. distance from the droplet center (s.d., in dashed lines). **j** Ratio of mean normalized intensities for NPM1-A488 and S6N-647 (black trace, left axis) plotted vs. the distance from the droplet center showing an NPM1-enriched outer layer. The mean intensity values from (**i**) are shown for reference (right axis). **g–j**, $n = 30$ radii

nM; Supplementary Fig. 13) than the IDR has for itself ($K_D \approx$ hundreds of micromolar[10]). Interestingly, S6N begins to invade the NPM1 network, partially replacing NPM1, only 200 min after the addition of excess S6N (Fig. 5c, d). We speculate that this may be a consequence of kinetic and thermodynamic barriers to invasion of the aged, predominantly homotypic NPM1 scaffold by S6N molecules. However, after displacement of ~10% of the NPM1 molecules, the increasingly heterotypic scaffold becomes more receptive to additional S6N molecules. Together, the results of these scaffold rearrangement experiments suggest that the homotypic (NPM1-NPM1) and heterotypic (NPM1-S6N) scaffolds form miscible, co-existing liquid phases that dynamically interconvert in response to changes in the protein composition of the surrounding milieu.

**An energetically favored NPM1-rich shell surrounds droplets**. To develop further support for our model of miscible, co-existing homotypic and heterotypic phases within NPM1-S6N droplets, we next examined the material properties of liquid-phase scaffolds with two extreme compositions (both with 5% PEG), pure homotypic NPM1 droplets and predominantly heterotypic droplets comprised of 4:1 S6N:NPM1. First, we measured the viscosity of the two types of droplets using the fluorescent molecular rotor DCVJ, whose quantum yield is viscosity-dependent[29]. We determined that the pure homotypic and predominantly heterotypic droplets exhibit similar viscosity values of 1.8 ± 0.1 and 2.6 ± 0.2 Pa·s, respectively (Fig. 5e and Supplementary Fig. 14). Next, we qualitatively determined the wetting properties of each of the two types of droplets, on surfaces treated with a hydrophobic (SigmaCote) or a hydrophilic (Pluronic F-127) agent, to qualitatively compare the hydrophilic/hydrophobic character of the two NPM1-based scaffolds. While the two scaffolding blends exhibited similar wetting properties on the hydrophilic surface, the 4:1 S6N:NPM1 droplets exhibited a stronger preference for wetting the hydrophobic surface compared to the homotypic NPM1-NPM1 droplets (Fig. 5f), suggesting that in a hydrophilic milieu, the homotypic scaffold affords a lower interfacial energy compared to the more hydrophobic, heterotypic scaffold. High-resolution, 3D point scanning microscopy imaging revealed color separation at the droplet–buffer interface, indicating that this scaffold is enriched in NPM1 (Fig. 5g). We analyzed the spatial heterogeneity in composition within NPM1-S6N droplets using a custom image analysis algorithm termed Sauron. Sauron radially normalizes the fluorescence intensities for both NPM1 and S6N from the droplet center to the boundary (Fig. 5h), and then averages these values (Fig. 5i). To determine compositional variations from the center to the boundary of droplets, we calculated

the ratio of normalized radial average fluorescence intensity values for NPM1 and S6N (Fig. 5j, black trace). We found that the boundary of the droplet was apparently enriched with NPM1 compared to the interior (Fig. 5j). This inhomogeneity in compositions was also evident in reconstructed Z-stack images of NPM1-S6N droplets (Supplementary Fig. 15A, B). To validate that this was not an artifact from chromatic aberrations, we imaged homogeneously fluorescent beads of comparable size using identical acquisition parameters. We show that a notable heterogeneity exists within the droplet compared to the standard beads where the NPM1:S6N normalized fluorescence intensity ratio increases with increasing distance from the slide surface (Supplementary Fig. 15C, D). Furthermore, we consistently observed this heterogeneity on both hydrophobic and hydrophilic surfaces, suggesting that it was not due to preferential interaction with the slide surface (Supplementary Fig. 15B, D). We further tested NPM1-S6N droplets with compositions ranging from S6N: NPM1 molar ratios of 5:1 to 0.25:1. All of these droplets exhibited an NPM1-rich layer (compared to the interior of the droplet; Supplementary Fig. 16) of similar width at the interface with buffer. Notably, the varied S6N:NPM1 molar ratios were also reflected in the composition of the interfacial layer. Together, these observations suggest that the heterotypic droplets adopt a core-shell architecture, wherein an NPM1-rich shell minimizes the interfacial energy with the aqueous solvent.

## Discussion

The late-processing steps of the ribosome biogenesis pathway, which involves stoichiometric assembly of ribosomal proteins with rRNA, occur in the outer, liquid-like GC layer of the nucleolus prior to the exit of pre-ribosomal particles into the nucleoplasm[3]. Interestingly, cells can sense aberrant nucleolar function. For example, impaired ribosome production causes ribosomal proteins such as rpL5, rpL11, and rpL23 to exit the nucleolus, and bind to and inhibit the E3-ligase activity of Mdm2, which in turn activates the tumor suppressor p53 and its downstream effectors, thereby inducing cell cycle arrest or apoptosis[30]. This is termed the ribosomal protein stress response. In addition to ribosomal proteins, the GC is enriched in RNA-binding, non-ribosomal proteins, such as NPM1, SURF6, Ki-67, and others[3,14]. Selective knockdown of non-ribosomal, nucleolar proteins often results in disrupted nucleolar morphology and cell cycle arrest[12,31–33], suggesting that alterations in non-ribosomal protein levels affect ribosome assembly and trigger the ribosomal protein stress response. Several questions arise from these observations: (1) How does the cohesive, liquid-like micro-environment of the nucleolus, which arises from the collective

behavior of rRNA, ribosomal, and non-ribosomal proteins, enable signal transduction between the nucleolus and the nucleoplasm? And, (2) How is the directionality of ribosomal biogenesis, with ribosomal particles assembled from the inner to outer region of the GC, orchestrated within this liquid-like microenvironment?

We previously proposed that the proteins and rRNAs localized within the GC of the nucleolus dynamically intermingle within extended molecular networks formed through weak, multivalent interactions of two types: protein–protein interactions involving electrostatically complementary acidic tracts and R-motifs (e.g., as found in NPM1 and SURF6, respectively; Supplementary Fig. 17A, B), and protein–RNA interactions involving either folded or disordered RNA-binding domains and rRNA[9]. Homotypic interactions between RGG domains[11,32] (e.g., as found in Nucleolin and FIB; Supplementary Fig. 17C, D) and between alternating acidic and basic tracts within the same polypeptide chain (e.g., as found in NPM1 and Nucleolin; Supplementary Fig. 17A, C) likely mediate inter-protein contacts that contribute to extended inter-molecular networks that underlie phase separation within the GC. These complex macromolecular networks, which also involve heterotypic protein–rRNA contacts, provide a liquid-like scaffold that is conducive to ribosome assembly and, when disrupted, releases signaling factors (e.g., rpL5, rpL11, and rpL23[30]) that activate cell stress response pathways.

Here, we demonstrate a role for the non-ribosomal protein SURF6 in modulating the accessible valency of NPM1 and the NPM1-dependent molecular scaffold in liquid-like droplets. While its functional role in the nucleolus is unknown, SURF6 is evolutionarily conserved, with homologs identified even in yeast[34], in contrast to NPM1 which evolved later[35]. Similar to NPM1[36], high SURF6 levels are associated with cellular hyperproliferation and enhanced ribosome biogenesis[34], supporting the hypothesis that NPM1 and SURF6 jointly contribute to the formation and functional regulation of the GC scaffold. Further, the expression of SURF6 in activated lymphocytes is delayed with respect to that of NPM1[37], suggesting that temporal regulation of SURF6 may play a role in lymphocytes by modulating the liquid-like features of the GC scaffold.

Similar to NPM1, SURF6 also interacts with nucleic acids (DNA and RNA)[12], as well as other abundant nucleolar proteins, such as the RNA Pol I upstream binding factor, UBF1 (localized to the FC), and Nucleolin (localized to the DFC and GC)[38]. Interestingly, UBF1 and Nucleolin also exhibit long acidic tracts (Supplementary Fig. 17 C, E), similar to NPM1, and thus are likely to bind to SURF6 through the same type of electrostatic interactions. Based on these observations, we propose that the role of SURF6 in regulating the composition and biophysical properties of the nucleolar matrix extends beyond the GC, and into the FC and DFC.

Through a diverse array of competitive interactions with multiple nucleolar proteins, as well as DNA and RNA, SURF6 may dynamically modulate the features of the nucleolar scaffold during ribosome biogenesis, possibly contributing to the formation of a nucleolar scaffolding gradient that directs the path of ribosomal particle assembly. A model for how NPM1's multiple mechanisms of LLPS may mediate vectorial ribosomal subunit assembly was previously discussed[10]. Here, we show that incorporation of SURF6 within homotypic NPM1 droplets mobilized NPM1 that was previously highly immobile (Fig. 4c), and altered droplet viscosity (Fig. 5e), composition (Fig. 5d), and hydrophobicity (Fig. 5f). Thus, discontinuity of the SURF6 concentration within the nucleolus, and the associated effects on scaffold viscosity and hydrophobicity, could contribute to this hypothetical ribosome assembly-promoting gradient. Interestingly, dramatic differences in local viscosity, hydrophobicity, and surface

tension were shown to mediate the compartmentalization of the DFC inside the GC[11]. Future studies will be required, however, to test our hypotheses regarding how the different types of competitive scaffolds, involving numerous proteins and nucleic acids, influence the molecular rearrangements within the nucleolus that accompany vectorial ribosome biogenesis.

## Methods

**Protein expression and purification.** Recombinant poly-histidine-tagged full-length NPM1- and GST-tagged ΔN-NPM1 constructs in pET28a (+) (Novagen) and pGEX-6p-3 (GE Healthcare) plasmids, respectively, were expressed in BL21 (DE3) *Escherichia coli* cells (Millipore Sigma, Burlington, MA, USA)[9]. Full-length NPM1 and ΔN-NPM1 were purified from the soluble fraction using Ni-NTA and glutathione-agarose affinity chromatography, respectively. Affinity tags were removed via proteolytic cleavage with TEV (for NPM1) and Turbo3C (for ΔN-NPM1; BioVision, Milpitas, CA, USA) protease, and passed through a C4 HPLC column (Higgins Analytical, Mountain View, CA, USA) for final purification. NPM1 constructs were refolded by resuspending lyophilized proteins in 6 M guanidinium HCl and dialyzing overnight against 10 mM Tris, 150 mM NaCl, 2 mM dithiothreitol (DTT), pH 7.5 buffer. Aliquots of NPM1 constructs were flash frozen and stored at −80 °C.

Poly His-tagged S6N containing a TEV cleavage site were expressed in BL21 Rosetta 2 (DE3) *E. coli* cells (Millipore Sigma, Burlington, MA, USA) and purified from inclusion bodies[10]. Cells were lysed by resuspension in 50 mM sodium phosphate 0.1% Triton X-100, pH 8.0. Pellet containing His-S6N was collected by centrifugation and dissolved in 6 M guanidinium HCl. Affinity purification was performed using a Ni-NTA column under denaturing conditions. Purified His-tagged S6N was dialyzed against 20 mM sodium phosphate 200 mM NaCl, 2 mM DTT, 2 mM EDTA, pH 8.0, and treated with TEV protease to remove the tag. Cleaved S6N proteins were passed through a C4 HPLC column (Higgins Analytical, Mountain View, CA, USA) for final purification. Lyophilized proteins were re-suspended in 6 M guanidine HCl and dialyzed in high salt buffer (10 mM Tris, 1 M NaCl, 2 mM DTT, pH 7.5). The proteins were stored in the 1 M NaCl buffer as at −80 °C. Final salt concentration used in the study (150 mM NaCl) was achieved by dilution with 10 mM Tris, 2 mM DTT, pH 7.5.

**Turbidity assays.** Phase separation of NPM1 and NPM1-S6N was determined by monitoring solution turbidity in the presence of PEG (MW = 8 kDa) in 10 mM Tris, 150 mM NaCl, 2 mM DTT, pH 7.5. To construct the phase diagram for NPM1-S6N solutions, turbidity was monitored over a wide range of NPM1 and S6N concentrations and PEG concentrations. Ten microliters of NPM1-S6N samples were prepared by mixing S6N in solutions containing crowding agents and NPM1 in buffer. Samples were incubated for 10 min at room temperature and vortexed prior to measuring the absorbance at 340 nm using a NanoDrop 2000c spectrophotometer (Thermo Scientific, Waltham, MA, USA). LLPS was also tested in the presence of other crowding agents by mixing 10 μM NPM1 and 10 μM S6N with 15% dextran sulfate (MW = 10 kDa) or Ficoll (MW = 70 kDa). Measurements were performed in triplicate. Solutions were scored positive for LLPS when $A_{340} \geq 0.1$.

**Fluorescent labeling of proteins.** NPM1 and S6N proteins were fluorescently labeled using maleimide derivatives of Alexa Fluor 488 and Alexa Fluor 647 (Thermo Fisher Scientific, Waltham, MA, USA) respectively, according to the manufacturer's manual[9,10]. NPM1 was conjugated with Alexa Fluor 488 at Cys104 (NPM1-A488). S6N was labeled with Alexa Fluor 647 at Cys19 (S6N-A647). To generate NPM1 pentamers labeled at a single subunit, fluorescently labeled NPM1-A488 monomers were mixed with unlabeled NPM1 monomers at 1:9 ratio in 6 M guanidine HCl and refolded in 10 mM Tris, 150 mM NaCl, 2 mM DTT, pH 7.5, by dialysis.

**Quantitation of NPM1:S6N ratios inside droplets.** To determine the effect of crowding on compositions of NPM1-S6N droplets, NPM1 (10% NPM1-A488) was mixed with S6N (10% S6N-A647) pre-equilibrated with PEG in 10 mM Tris, 150 mM NaCl, 2 mM DTT, pH 7.5. The final concentrations of NPM1 and S6N are 10 μM. At 5% PEG, the order of addition of NPM1, S6N, and PEG did not alter the ratio of components inside droplets. However, at 15% PEG, pre-mixing of NPM1 and S6N was required prior to addition of PEG to produce droplets with homogeneous compositions. Droplet solutions were transferred on 16-well CultureWell chambered slides (Grace BioLabs, Bend, OR, USA) coated with PlusOne Repel Silane ES (GE Healthcare, Pittsburgh, PA, USA) and Pluronic F-127 (Sigma-Aldrich, St. Louis, MO, USA) and incubated for 4 h. Images were acquired using Zeiss LSM 780 NLO point scanning confocal microscope (Carl Zeiss Microscopy GmbH, Jena, Germany) with ×63 Plan Apochromat (N.A. 1.4) objective. To correlate the fluorescence intensities of droplets derived from the images, calibration curves from images of free Alexa Fluor 488 and Alexa Fluor 647 dye solutions were generated. Calibration plots were constructed from mean fluorescence intensities for the entire field of view of microscopy images of dye solutions with known concentrations (Supplementary Fig 2). In order to confirm that the fluorescence

intensity measured within droplets reports directly on the protein concentration and is not convoluted with other potential photophysical artifacts (i.e., auto-FRET, self-quenching, etc.), we measured the fluorescence intensity of the dense phase in phase separated samples formed with 0.5–10% labeled protein. Data in Supplementary Fig. 3 validated that the fluorescence signals within droplets vary linearly with the labeled protein concentrations under the experimental conditions used in this manuscript (10% labeled, 90% unlabeled protein). Droplet images were acquired using the same parameters used for the free dye solutions. The total protein concentrations inside the droplets were derived from the concentration of labeled proteins based on the mean intensities of droplets multiplied by the dilution factors to account for the fraction of labeled proteins in the mixtures. The concentrations were adjusted for the effect of high viscosities on the quantum yields of the fluorescent dyes by dividing with appropriate correction factors. The correction factors used for Alexa Fluor 488 and Alexa Fluor 647 were 0.73 and 1.43, respectively[10]. All analyses were performed using Fiji image processing software[39]. A similar analysis was performed for the quantification of NPM1 and S6N concentrations inside droplets under different crowding conditions. In this case, standard curves for the specific crowder concentrations were used.

**Fluorescence recovery after photobleaching**. FRAP experiments on NPM1 and NPM1-S6N droplets were performed using 3i Marianas spinning disk confocal microscope (Intelligent Imaging Innovations Inc., Denver, CO, USA) with a 100X oil immersion objective (N.A. 1.4). Heterotypic droplets (10 μM NPM1, 10 μM S6N) spiked with NPM1-A488 were prepared in 10 mM Tris, 150 mM NaCl, 2 mM DTT, pH 7.5 buffer with or without 5% or 15% crowding agents and transferred to Repel Silane/PF-127-coated chambered slides. A circular area (diameter = 1μm) near the center of the droplet was photobleached to at least 50% of initial fluorescence intensities. Recovery of fluorescence was monitored every ~500 ms. FRAP for multiple droplets were processed using Slidebook 6.0 (Intelligent Imaging Innovations, Gottingen, Germany). Intensities were normalized and corrected for global photobleaching during image acquisition. FRAP recovery curves were fitted to determine the recovery times and mobile fractions according to Eq. 1:[40]

$$I_t = \frac{I_0 + I_\infty \frac{t}{t_{1/2}}}{1 + \frac{t}{t_{1/2}}}, \qquad (1)$$

where $I_t$ is the fluorescence intensity at time point $t$, $I_\infty$ is the maximum intensity value after bleaching, $I_0$ the intensity immediately after bleaching, and $t_{1/2}$ is the time required for half of the total fluorescence intensity to be recover. The apparent diffusion coefficients were extracted from Eq. 2:[41]

$$D_{app} = 0.224 r^2 / t_{1/2}, \qquad (2)$$

where $r$ is the radius of the bleached, circular area. FRAP experiments performed on heterotypic droplets in 5% PEG at various S6N-NPM1 ratios with NPM1 concentration set at 5 μM were done similarly. FRAP for NPM1-A488 and S6N-A647 were performed on separate sets of droplets. For homotypic and heterotypic droplet aging experiments, FRAP were performed at different time points after mixing of 20 μM NPM1 in 5% PEG, 10 mM Tris, 150 mM NaCl, 2 mM DTT, pH 7.5 buffer.

**Determination of NPM1 concentrations in the light phase**. To determine the concentrations of NPM1 in the light phase of homotypic droplets formed in different PEG concentrations, we used fluorescence spectroscopy. Thirty microliters of NPM1 solutions (10 μM NPM1 with 16% NPM1-A488) with different concentrations of PEG (0%, 5%, 15%, and 30%) were prepared and centrifuged to separate the dense and light phases. Ten microliters of the light phase were transferred into 384-well microplate (Greiner Bio-One International, Kremsmünster, Austria). Fluorescence emission at 520 nm was measured at 25 °C using a CLARIOstar microplate reader (BMG Labtech, Ortenberg, Germany) after excitation at 472 nm. To estimate NPM1-488 concentrations, fluorescence measurements were performed on standard solutions of Alexa-488 dye in different percentages of PEG to generate calibration plots. Total NPM1 concentrations were determined by dividing the concentrations of NPM1-A488 by the fraction of NPM1-A488 in solution.

**Determination of S6N incorporation into NPM1 droplets**. Aged NPM1 droplets were prepared by mixing 10 μM NPM1 (10% NPM1-A488) with 5% PEG, 10 mM Tris, 150 mM NaCl, 2 mM DTT, pH 7.5 buffer, and incubated overnight at room temperature on a chamber slide. Droplets prior to S6N addition were imaged with a Zeiss LSM 780 NLO point scanning confocal microscope. S6N (10% S6N-A647) was added to a final concentration of 5 μM. Images were acquired at 10 min time intervals over 18 h. To determine displacement of NPM1 from droplets, NPM1-S6N droplets were prepared by adding 5 μM S6N to aged NPM1 (prepared as described above) and incubated overnight, at room temperature. Pre-incubation of NPM1 with sub-stoichiometric concentrations of S6N before adding excess S6N prevented formation of several new droplets that interfere with time-lapse imaging. Additional S6N (35 μM) was mixed to obtain the final concentration of 40 μM. Images were acquired overnight. Correction factors for photobleaching of NPM1-

A488 and S6N-A647 during the entire image acquisition were applied. The rate of photobleaching under the same acquisition conditions were determined for NPM1-A488 in homotypic NPM1 droplets and for S6N-A647 in NPM1-S6N heterotypic droplets (10 μM each), in 5% PEG. The photobleaching curves were fitted to a single exponential equation to determine the rate of bleaching for both labeled proteins[42]. NPM1-A488 intensities were normalized to the mean intensities of NPM1 droplets from images acquired immediately after the addition of S6N.

**Analysis of NPM1-enriched droplet/solvent interface**. Fluorescently labeled NPM1-S6N droplets were prepared by mixing 10μM NPM1 (10% NPM1-A488) and 10μM S6N (10% S6N-A647) with 5% PEG, 10 mM Tris, 150 mM NaCl, 2 mM DTT, pH 7.5 buffer, and transferred to chambered slides. Z-stacks of 2D images were collected using a Zeiss LSM 780 NLO point scanning confocal microscope with 63X oil immersion objective (NA 1.4). Z-stack images were acquired to allow analysis of droplets across a range of XY planes. We have established that the heterogeneity in the fluorescence distributions for NPM1-A488 and S6N-A647 were not artifacts from chromatic aberrations by imaging standard homogeneously fluorescent 4-μm beads (Thermo Fisher Scientific, Waltham, MA, USA) using the same imaging parameters as used for the droplets. Intensities for both channels superimpose with homogeneous intensity ratios throughout, confirming that the observed NPM1-enriched interface is not an optical aberration.

To quantitatively assess the presence of NPM1-enriched interface we used Sauron, an algorithm we developed that radially averages the normalized fluorescence intensities derived from NPM1-A488 (green channel) and S6N-647 (red channel) in a droplet across XY planes. Briefly, this algorithm is divided into three portions: Sauron, Saruman, and Grima. Sauron calls Saruman on each slice so that the analysis and normalization is performed on each slice independently. Saruman normalizes the intensities on a given Z-slice, such that all values for red and green channels are in [0,1], and continues to find droplets (as 2D circles) by segmentation and validation of circularity, while the maximum pixel intensities for unanalyzed droplets are above a given threshold (75% of normalized NPM1-A488 intensity for the slice). When Saruman has performed the segmentation and validation on a droplet slice, the slice is used as input to Grima (which performs a secondary normalization on the segmented droplet slice) to perform the radial averaging of intensities on the green and red channel.

**Determination of droplet viscosity**. To generate homotypic droplets, 20 μM NPM1 was mixed with 5% PEG in 10 mM Tris, 150 mM NaCl, 2 mM DTT, pH 7.5, containing 10 μM 9-(2,2-dicyanovinyl)julolidine (DCVJ; Sigma-Aldrich, St Louis, MO, USA). Heterotypic droplets were prepared by mixing 5 μM NPM1 with 20 μM S6N in 5% PEG in 10 mM Tris, 150 mM NaCl, 2 mM DTT, pH 7.5, containing 10 μM DCVJ. Images of droplets were acquired using the Zeiss LSM 780 NLO point scanning confocal microscope at ($\lambda_{ex}$ = 488 nm; $\lambda_{em}$ = 562 nm). A standard curve using glycerol/water solutions of different mass fractions (http://www.met.reading.ac.uk/~sws04cdw/viscosity_calc.html) were generated to correlate DCVJ fluorescence intensities with viscosity. Glycerol/water solutions were mixed with 10μM DCVJ and imaged using the same parameters as with the droplets.

All reported experiments in this manuscript were performed in triplicate, at a minimum.

**Code availability**. Sauron is written in Python 2.7.2 and the source code is available for download here: https://github.com/drjrm3/sauron.

## Data availability
Other data are available from the corresponding author upon reasonable request. A Reporting Summary for this Article is available as a Supplementary Information file.

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

## Acknowledgements

We thank Dr. Christopher Stanley at Oak Ridge National Laboratories for assistance with SAXS data collection for S6N and Mr. Ramiz Somjee for assistance with Supplementary Fig. 17. Images were acquired at the Cell & Tissue Imaging Center which is supported by St. Jude Children's Research Hospital and NCI P30 CA021765; we thank Dr. Victoria Frohlich and Dr. Sharon King for the technical assistance. This work was supported by National Institutes of Health grant R01GM115634 to R.W.K., National Cancer Institute grant CA021765 to St. Jude Children's Research Hospital, the St. Jude Collaborative Research Consortium on Membraneless Organelles, and ALSAC.

## Author contributions

M.C.F. designed the study, performed experiments, interpreted data, and wrote the paper; D.M.M. designed the study, interpreted data, and wrote the paper; J.R.M. developed the Sauron algorithm; R.W.K. designed the study, interpreted data, and wrote the paper.

## Additional information

**Competing interests:** The authors declare no competing interests.

