## [Peer Review File · Nature Communications]

Reviewers' comments:

Reviewer #1 (Remarks to the Author):

The nucleolus is a multi-functional, multi-component membraneless organelle that is the site of assembly of ribosomal subunits and also serves as depot for sequestering key stress response proteins that enable the shutting down of ribosomal biogenesis during stress. The functions of nucleoli are thought to be governed by the layered architecture of this organelle. Further, this layered architecture is thought to display modulated phases, which refers to the differential fluxes of ribosomal and non-ribosomal proteins across the different layers. The authors propose a simple ansatz in this work: The homotypic interactions amongst the main component - NPM1 - of the outer layer (the GC) should lead to a percolated network of NPM1 molecules. Fluxing of proteins that compete for interactions with modules along NPM1 will shear the network thus enabling the influx of ribosomal proteins and the efflux of assembled ribosomal subunits. This is an elegant and simple model for describing the synergy between phase behavior of nucleolar components and functions of nucleoli.

In this work, Kriwacki, Mitrea and coworkers test the tenets of their model, proposed in earlier work. To do this, they quantify the phase behavior of NPM1 mediated by "homotypic" interactions and the effects of competing heterotypic interactions on the homotypic phase behavior. This is a very interesting study and it is likely to set the stage for a comprehensive model that captures the synergies between spontaneous and driven processes. The manuscript deserves to be published. However, there is a major conceptual issue that needs to be clarified and several revisions that need to be made in order to ensure that the narrative is up to date.

Major concern

The so-called homotypic phase behavior is thought to be driven by complementary acidic and basic tracts within the IDR of NPM1. This seems reasonable. However, these interactions take effect only in the presence of PEG. Here comes the challenge. PEG is thought to enhance homotypic interactions (strengthen them and enable increased physical crosslinking) through what in the physical literature will be known as the depletion effect and in the biophysical literature is an effective increase in NPM1 concentration through an excluded volume effect. Implicit through the manuscript is the hypothesis that PEG is an inert crowder that strengthens homotypic interactions among NPM1 molecules through an effective concentration mechanism. The authors show that increasing the weight percent of PEG leads to more concentrated droplets. This is where the narrative becomes conceptually tricky. The system is closed and the volume fraction of protein is a conserved order parameter. So an increase in the concentration of NPM1 within the droplet has to be accompanied by a decrease in the concentration of NPM1 in the light phase. Figure 2a seems to suggest that the concentration of NPM1 in the light phase remains unchanged with increased weight percent of PEG. This would imply that the slope of the tie line changes as the PEG weight percent changes and that there is PEG being partitioned into the droplets. PEG, in this scenario, cannot be inert. Instead, it might serve as a scaffold for enabling the crosslinking of NPM1 molecules. This inference is supported by the odd order of operations dependence of droplet formation in the presence of 15% PEG as opposed to 5% PEG. The key questions are as follows: Are the interactions truly homotypic? Is PEG excluded from droplets? If not, then how much of the PEG is within droplets? And if PEG is present within droplets, are the interactions truly homotypic? To answer these questions, there are few key experiments that need to be added: Measure the concentration of NPM1 in the light phase in the presence of increasing weight percents of PEG. Do these concentrations increase, decrease, or stay the same with the PEG weight percent? If they decrease, then the model being suggested has merit. If it stays the same, then there is partitioning of PEG into the droplets, but the increased concentration of NPM1 within the droplets is coming from the preferential exclusion of PEG from the droplets, i.e., the concentration of PEG being higher in the light phase leads to equalization of the chemical potential across the phase boundary. If the NPM1 concentration in the light phase increases with increasing PEG weight percent, then there is a preferential partitioning of PEG into the droplets. In either of the latter two

scenarios, PEG is not inert and it is exerting influence on the phase behavior through linked equilibria. The outcome of this experiment, i.e., a precise measurement of the NPM1 concentration in the light phase for different PEG weight percents, will dictate the precise model that applies and allows us to understand the role of either preferential interactions or scaffolding effects of PEG. To iron out these possibilities, an additional experiment would be essential: Increase the molecular weight of PEG and adjust the weight percent to match the weight percent being used for the current MW and ask if the phase behavior one obtains is equivalent. If there is a MW dependence, then PEG is not inert and nor would dextran or ficol. These are polymers with functional groups. They aren't inert, hard obstacles. In fact, for inert hard obstacles, a strong case can be made for the suppression of phase separation by quenching fluctuations that need to grow.

A second, potentially thorny issue is the increase of the weight percent of PEG without accounting for non-idealities. The osmotic pressure in PEG:water mixtures increases as the $9/4$ power of PEG concentration past the overlap concentration. This leads to interesting non-idealities due to PEG at higher weight percents. Indeed, it might point to the fact that the solvent in the droplet is fundamentally different from the solvent in the light phase, partly due to the contributions from PEG.

The preceding points are super important because either the authors are absolutely right and they are observing a stabilization of homotypically driven phase separation via an inert crowder (which ratifies the pursuit of their model) or there are linkage effects due to the presence of the crowder that are entirely non-trivial. This scenario would give one pause for embracing the model proposed by the authors because the enhanced homotypic interactions may be rather non-trivial.

Minor points

1) Figure 2A is impossible to parse. The colored symbols are not explained in the caption and given the importance of the light phase concentrations a more clear version of this figure is absolutely essential. It appears also that panel C should be D and vice versa if the figure is go with the caption.

2) In Figure 3C, should the figure for 5% PEG be for 15% PEG and vice versa? Otherwise, this becomes a strange and impossible to understand result.

3) Alex-488 can self-quench. The same is true, albeit to a lesser extent for Alexa-647. This is important because it relates directly to the inferred concentrations of NPM1 and SURF-6 within the droplets. This self-quenching would need to be accounted for in addition to the viscosity correction. One way around this is to query the robustness of the inferred concentrations when using 5% or 1% labeled molecules instead of 10%. Also, the authors are assuming a homogeneous distribution of fluorophores when applying their correction factors. How good is this assumption?

4) The authors seem surprised that fibrils don't form within NPM1 droplets. It is not clear why this should be surprising. It is noteworthy that in the presence of the full complement of interactions - as opposed to those of the LCD alone - FUS can go through a cascade of interactions involving a synergy between the Tyr-rich PLD and Arg-rich RBD. See <http://doi.org/10.1016/j.cell.2018.06.006>, which calls into question the relevance of fibrils formed by the PLD alone. Even the systems that Rosen studies are poised to undergo gelation transitions based on the extent of crosslinking. In fact, the observations made in the current MS are resonant with the theoretical / computational work of Harmon et al. <https://elifesciences.org/articles/30294>. It would be constructive to revise the section that discusses gelation with / without fibril formation.

5) Finally, I found the title to be obtuse. It is not clear that the average reader will readily appreciate what this paper is about from the title. Something that's more direct and self-explanatory would be helpful to the authors' cause.

I conclude by noting that the source of the result in Figure 3A was non-obvious to me. One can write out all the possible mathematical scenarios and this leads to clear expectations for the concentration of NPM1 in the light phase, the slopes of tie lines, and the partitioning of PEG into / out of the droplet. It is based on this analysis that I furnish the predictions laid out above. Data speak for themselves. The key point is that the support for the model being articulated here is incomplete absent the measurement of [NPM1] in the light phase and investigating the coupling between increase / decrease / stasis of this concentration vis-a-vis the data shown in Figure 3A. Either the authors are absolutely right and crowders influence the strengths of homotypic interactions via depletion effects and nothing else (i.e., the crowders are inert) or there is selective partitioning of PEG within the droplets and this contributes to the crosslinking as well as modulation of the affinities between acidic and basic tracts.

No matter the outcome of the clarifying experiments, this is likely to be a highly influential and important paper. It should be published, albeit with a clear assessment of what PEG is actually doing.

Reviewer #2 (Remarks to the Author):

Ferrolino et al. studies different aspects of co-phase separation of two nucleolar proteins: NPM-1 and SURF6, including the effect of molecular crowding, measurement of material properties, the influence of SURF-6 on the concentration and dynamics of NPM-1 within phase-separated condensates. The authors convincingly show that phase separation dependent on homotypic NPM-1-NPM-1 interaction generates condensates that differ in composition and dynamics from condensates with both homotypic NPM-1-NPM-1 and heterotypic NPM-1-SURF-6 interactions.

Co-phase separation involving heterotypic interactions between two proteins or between a protein and an RNA have been well studied in other systems (Banani et. al. Cell 2016, Zhang et. al. Mol Cell 2015 for example). This study with the nucleolar proteins does not offer significantly novel insights in terms of broadly thinking about co-phase separation of two macromolecules.

The authors propose that some aspects of co-phase separation between NPM-1 and SURF-6 can control the directionality of ribosome biosynthesis within the nucleolus. This claim is too preliminary and needs to be substantiated with significant *in vivo* and *in vitro* experiments.

Major concerns

1) How crowding affects dynamics in condensates needs to be clarified. A) The authors claim that PEG crowding works through volume exclusion effects. Does PEG concentrate within the condensates? How does phase separation respond to crowding with PEG of different sizes? B) The authors claim that PEG crowding increases NPM-1-NPM-1 interactions within homotypic droplets and stabilization of these interactions over time leads to maturation of condensates. Why is maturation faster at lower concentration of PEG (see Figure 3C)? Does PEG play a more significant role rather than being a passive crowding agent?

2) The conclusion that condensates containing both NPM-1 and SURF-6 have an NPM-1 rich shell (boundary) is not convincing. In the condensate presented in Figure 5G, while the right-side boundary looks green, the left-side boundary looks yellow. In the quantification presented in Figure 5J, NPM-1 appears to be only ~50% more enriched at the interface compared to SURF-6. Have the authors considered the possibility that the interface contains about the same number of molecules of NPM-1 and SURF-6, while the observed 50% difference is due to the use of different fluorophores to label NPM-1 and SURF-6. The figure legends for 5I-J should describe more completely what the reaction mix contains.

3) Could the authors repeat some of their experiments with a mutant of SURF-6 or NPM-1 that

weakens/strengthens the interaction between the two proteins? This will help strengthen the claim that the observed differences are indeed due to binding between SURF-6 and NPM-1.

Minor concerns

1) The word "cross-linking" is often used to describe "interactions" in condensates. Unless there is evidence that crosslinking occurs, the word should be revised.

2) The text often reads as if S6N is a scaffold while it is actually a client. The scaffold NPM-1 comes in two forms – in complex with itself or with S6N. Revision could be considered.

3) Figures 2, 4: The authors should clarify in the figure itself or figure-legend about the configuration of the FRAP experiment – i.e. a small internal area bleached to probe for internal rearrangement of macromolecules vs. bleaching a whole condensate to additionally monitor exchange of macromolecules between condensate and surroundings.

4) Are there any estimates available regarding the viscosity of nucleolus granular component in cells? If yes, how does it compare with the values measured here by the authors?

Point-by-point responses to Reviewers' comments

We thank the reviewers for their time and effort in evaluating our manuscript and for their suggestions for improving its clarity and quality.

Responses to Reviewer #1

Reviewer 1, point 1: *“The nucleolus is a multi-functional, multi-component membraneless organelle that is the site of assembly of ribosomal subunits and also serves as depot for sequestering key stress response proteins that enable the shutting down of ribosomal biogenesis during stress. The functions of nucleoli are thought to be governed by the layered architecture of this organelle. Further, this layered architecture is thought to display modulated phases, which refers to the differential fluxes of ribosomal and non-ribosomal proteins across the different layers. The authors propose a simple ansatz in this work: The homotypic interactions amongst the main component - NPM1 - of the outer layer (the GC) should lead to a percolated network of NPM1 molecules. Fluxing of proteins that compete for interactions with modules along NPM1 will shear the network thus enabling the influx of ribosomal proteins and the efflux of assembled ribosomal subunits. This is an elegant and simple model for describing the synergy between phase behavior of nucleolar components and functions of nucleoli.*

In this work, Kriwacki, Mitrea and coworkers test the tenets of their model, proposed in earlier work. To do this, they quantify the phase behavior of NPM1 mediated by "homotypic" interactions and the effects of competing heterotypic interactions on the homotypic phase behavior. This is a very interesting study and it is likely to set the stage for a comprehensive model that captures the synergies between spontaneous and driven processes. The manuscript deserves to be published. However, there is a major conceptual issue that needs to be clarified and several revisions that need to be made in order to ensure that the narrative is up to date.”

Authors: We thank the reviewer for their enthusiasm regarding our findings.

Reviewer 1 point 2: *“The so-called homotypic phase behavior is thought to be driven by complementary acidic and basic tracts within the IDR of NPM1. This seems reasonable. However, these interactions take effect only in the presence of PEG. Here comes the challenge. PEG is thought to enhance homotypic interactions (strengthen them and enable increased physical crosslinking) through what in the physical literature will be known as the depletion effect and in the biophysical literature is an effective increase in NPM1 concentration through an excluded volume effect. Implicit through the manuscript is the hypothesis that PEG is an inert crowder that strengthens homotypic interactions among NPM1 molecules through an effective concentration mechanism. The authors show that increasing the weight percent of PEG leads to more concentrated droplets. This is where the narrative becomes conceptually tricky. The system is closed and the volume fraction of protein is a conserved order parameter. So an increase in the concentration of NPM1 within the droplet has to be accompanied by a decrease in the concentration of NPM1 in the light phase. Figure 2a seems to suggest*

that the concentration of NPM1 in the light phase remains unchanged with increased weight percent of PEG. This would imply that the slope of the tie line changes as the PEG weight percent changes and that there is PEG being partitioned into the droplets. PEG, in this scenario, cannot be inert. Instead, it might serve as a scaffold for enabling the crosslinking of NPM1 molecules. This inference is supported by the odd order of operations dependence of droplet formation in the presence of 15% PEG as opposed to 5% PEG. The key questions are as follows: Are the interactions truly homotypic? Is PEG excluded from droplets? If not, then how much of the PEG is within droplets? And if PEG is present within droplets, are the interactions truly homotypic? To answer these questions, there are few key experiments that need to be added: Measure the concentration of NPM1 in the light phase in the presence of increasing weight percents of PEG. Do these concentrations increase, decrease, or stay the same with the PEG weight percent? If they decrease, then the model being suggested has merit. If it stays the same, then there is partitioning of PEG into the droplets, but the increased concentration of NPM1 within the droplets is coming from the preferential exclusion of PEG from the droplets, i.e., the concentration of PEG being higher in the light phase leads to equalization of the chemical potential across the phase boundary. If the NPM1 concentration in the light phase increases with increasing PEG weight percent, then there is a preferential partitioning of PEG into the droplets. In either of the latter two scenarios, PEG is not inert and it is exerting influence on the phase behavior through linked equilibria. The outcome of this experiment, i.e., a precise measurement of the NPM1 concentration in the light phase for different PEG weight percents, will dictate the precise model that applies and allows us to understand the role of either preferential interactions or scaffolding effects of PEG. To iron out these possibilities, an additional experiment would be essential: Increase the molecular weight of PEG and adjust the weight percent to match the weight percent being used for the current MW and ask if the phase behavior one obtains is equivalent. If there is a MW dependence, then PEG is not inert and nor would dextran or ficol. These are polymers with functional groups. They aren't inert, hard obstacles. In fact, for inert hard obstacles, a strong case can be made for the suppression of phase separation by quenching fluctuations that need to grow.”

Authors: We thank the reviewer for their thorough analysis of the multitude of mechanistic scenarios that could explain our data. We now include additional experimental data which supports the model where PEG acts as an inert crowder to promote NPM1 phase separation, as opposed to acting as an active scaffold. The additional supporting data included in the manuscript are as follows:

1. We measured the concentration of NPM1 in the light phase of homotypic NPM1-droplet suspensions formed in the presence of 5, 15 and 30% PEG-8K (Fig. 3A, black curve). As expected from the mass balance conservation model, where PEG does not contribute to scaffolding, the increase in [NPM1] within the dense phase observed with increasing percentages of PEG in solution is accompanied by a decrease in [NPM1] in the light phase.
2. Using a fluorescently labeled PEG-10K spiked into the crowded droplet suspensions, we demonstrated that PEG is distributed nearly uniformly within the light and dense phases of the homotypic NPM1 two-phase suspension

(Supplementary Fig. 5; Supplementary Table 2). For example, the partition coefficient for the labeled PEG-10K was 1.1 with 5% PEG-8K, and 1.3 with 15% and 30% PEG-8K. We consider this to be a minor extent of partitioning that is not a major factor in formation of the homotypic NPM1 droplets.

3. We performed turbidity assays of NPM1 in the presence of 15% PEG, while varying the PEG polymer chain length between 1,000 Da to 20,000 Da (Supplementary Fig. 6). The data show that while no effect is observed for the smallest polymer chain (PEG-1K), all other, longer polymer chains similarly promote phase separation of NPM1, regardless of polymer chain length (*i.e.*, PEG valency). Thus, the increased multivalency of the longer chains seems not to be a factor in the homotypic phase separation by NPM1 promoted by these polymers.

The following explanatory note is now included on page 6:

“Consistent with the increased concentration of molecules incorporated in the dense phase, light phase concentrations decreased (Fig. 3A; Supplementary Table 1). Two distinct mechanisms could be envisioned, that result in enhanced partitioning of NPM1 molecules within the dense phase. For one potential mechanism, PEG is an inert crowding agent, which causes chain compaction and stabilizes the electrostatic interactions between A- and B-tracts of NPM1 through volume exclusion effects, thereby driving the accumulation of NPM1 within the dense phase. For a second potential mechanism, PEG directly interacts with NPM1, and drives NPM1 accumulation within the dense phase by forming a heterotypic PEG-NPM1 scaffold. One fundamental feature of the NPM1 droplets that can discriminate between these two mechanisms is PEG partitioning. An inert crowding agent would be uniformly distributed between the light and dense phase (or be excluded from the dense phase if its dimensions are greater than the effective mesh size of the scaffold¹), while a scaffolding crowding agent would be enriched within the dense phase. Using Rhodamine B-labeled PEG-10K (PEG-RhB) as a probe, we measured partition coefficients between 1.1 ± 0.1 and 1.3 ± 0.1 within homotypic NPM1 droplets formed in the presence of 5%, 15% and 30% PEG, respectively (Supplementary Fig. 5; Supplementary Table 2), indicating that PEG molecules are freely diffusing between the two phases and do not favorably bind to NPM1. To further demonstrate that PEG is not part of the scaffold, we tested the PEG chain length dependence of LLPS. We compared NPM1 LLPS in solutions containing PEGs of different molecular weights (1 kDa, 4 kDa, 8 kDa and 20 kDa). We found that PEG-induced LLPS occurs at a minimum molecular weight of ~4 kDa, and larger PEGs do not further enhance phase separation (Supplementary Fig. 6). Interestingly, this range of PEG sizes (*e.g.*, >4 kDa) correlated with the previously reported long chain regime of crowders that can induce compaction of IDPs², suggesting that NPM1-IDR chain compaction may contribute to PEG-dependent phase separation. We conclude that PEG-dependent volume exclusion promotes NPM1-NPM1 interactions and homotypic LLPS, and may contribute to the immobilization of NPM1 within NPM1-S6N heterotypic droplets (Fig. 2C).”

Reviewer 1, point 3: “A second, potentially thorny issue is the increase of the weight percent of PEG without accounting for non-idealities. The osmotic pressure in

PEG:water mixtures increases as the 9/4 power of PEG concentration past the overlap concentration. This leads to interesting non-idealities due to PEG at higher weight percents. Indeed, it might point to the fact that the solvent in the droplet is fundamentally different from the solvent in the light phase, partly due to the contributions from PEG.

The preceding points are super important because either the authors are absolutely right and they are observing a stabilization of homotypically driven phase separation via an inert crowder (which ratifies the pursuit of their model) or there are linkage effects due to the presence of the crowder that are entirely non-trivial. This scenario would give one pause for embracing the model proposed by the authors because the enhanced homotypic interactions may be rather non-trivial.”

Authors: The new experimental data, presented now in Supplementary Fig. 5 and Supplementary Table 2, show that PEG is approximately equally distributed between the light and dense phases of NPM1 droplets, under all three PEG concentrations tested. As discussed above, these results indicate that PEG is an inert crowder. Therefore, we argue that consideration of non-idealities of PEG:water interactions are not pertinent to our key conclusion that crowding promotes NPM1:NPM1 interaction through volume exclusion. Respectfully, we prefer to not address this issue in the manuscript.

Reviewer 1, point 4: *“1) Figure 2A is impossible to parse. The colored symbols are not explained in the caption and given the importance of the light phase concentrations a more clear version of this figure is absolutely essential.”*

Authors: We thank the reviewer for noting confusing aspects of Fig. 2. To the first point, regarding the phase diagrams in Fig. 2A, we included an explanatory note in the figure caption, describing the meaning of the thin colored circles and their correlation with the microscopy images in panel 2B:

“(B) Confocal microscopy images of NPM1-A488 (green) and S6N-A647 (red) droplets in the presence of different PEG concentrations (0%, top; 5%, middle; and 15%, bottom); scale bar =10 μ m. NPM1 and S6N concentrations were 10 μ M each represented in the phase diagrams in (A) by thin blue, green and orange circles around the solid circles of similar color.”

Additionally, we modified the figure style to increase the contrast between the one phase and two-phase symbols.

Reviewer 1, point 5: *“It appears also that panel C should be D and vice versa if the figure is go with the caption.”*

Authors: We thank the reviewer for spotting the swapped labels. We apologize for this error and have fixed this in the caption of Fig. 2.

Reviewer 1, point 6: *“2) In Figure 3C, should the figure for 5% PEG be for 15% PEG and vice versa? Otherwise, this becomes a strange and impossible to understand*

result.”

Authors: The reviewer is correct to expect that the 15% PEG droplets should undergo gelation in a shorter time frame than the 5% PEG droplets. Upon re-examination in the primary data for these figure panels, we realized that there was high variability in the FRAP results for both the 5% and 15% PEG droplets. Consequently, we repeated the experiment with larger numbers of droplets and now include an updated, less noisy Fig. 3C and a corresponding updated Fig. 3B. The new data support the reviewer’s expectation. We thank the reviewer for drawing our attention to this issue.

Reviewer 1, point 7: “3) Alexa-488 can self-quench. The same is true, albeit to a lesser extent for Alexa-647. This is important because it relates directly to the inferred concentrations of NPM1 and SURF-6 within the droplets. This self-quenching would need to be accounted for in addition to the viscosity correction. One way around this is to query the robustness of the inferred concentrations when using 5% or 1% labeled molecules instead of 10%. Also, the authors are assuming a homogeneous distribution of fluorophores when applying their correction factors. How good is this assumption?”

Authors: We thank the reviewer for the cautionary note. We now include a control figure (Supplementary Fig. 3) which illustrates a linear correlation of fluorescence intensity inside droplets *versus* labeling percentage. In addition, we included the following details in the *Materials and Methods* section of the text:

(Page 17, bottom) “In order to confirm that the fluorescence intensity measured within droplets reports directly on the protein concentration and is not convoluted with other potential photophysical artifacts (i.e., auto-FRET, self-quenching, etc.), we measured the fluorescence intensity of the dense phase in phase separated samples formed with 0.5-10% labeled protein. Data in Supplementary Fig. 3 validated that the fluorescence signals within droplets vary linearly with the labeled protein concentrations under the experimental conditions used in this manuscript (10% labeled, 90% unlabeled protein).”

In order to ensure (or maximize) homogeneous distribution of fluorophores, when preparing our phase separated samples, we start with homogeneously pre-mixed, monodisperse labeled protein:unlabeled protein (1:9) solutions for NPM1 and S6N which are then combined to induce phase separation.

Reviewer 1, point 8: “4) The authors seem surprised that fibrils don’t form within NPM1 droplets. It is not clear why this should be surprising. It is noteworthy that in the presence of the full complement of interactions - as opposed to those of the LCD alone - FUS can go through a cascade of interactions involving a synergy between the Tyr-rich PLD and Arg-rich RBD. See <http://doi.org/10.1016/j.cell.2018.06.006>, which calls into question the relevance of fibrils formed by the PLD alone. Even the systems that Rosen studies are poised to undergo gelation transitions based on the extent of crosslinking. In fact, the observations made in the current MS are resonant with the theoretical / computational work of Harmon et al. <https://elifesciences.org/articles/30294>. It would be

constructive to revise the section that discusses gelation with / without fibril formation.”

Authors: We thank the reviewer for the comment. We softened the language on the unique nature of the reversibility of gelated NPM1 homotypic droplets. The paragraph on page 7 now reads:

“In contrast to several other phase separation-prone proteins associated with neurodegenerative diseases (e.g., FUS³ and hnRNPA1⁴), the gelated homotypic NPM1 droplets did not further transition to fibrillar structures after overnight incubation in the presence of 5% PEG. Instead, when the gelated droplets were re-suspended in buffer lacking crowding agent, they fully dissolved within eight hours (Fig. 3E-F).”

Reviewer 1, point 9: *“5) Finally, I found the title to be obtuse. It is not clear that the average reader will readily appreciate what this paper is about from the title. Something that’s more direct and self-explanatory would be helpful to the authors’ cause.”*

Authors: Yes, the original title was obtuse. Per reviewer’s suggestion, we changed the title to the following: “Compositional adaptability in NPM1-mediated scaffolding networks enabled by switching of phase separation mechanisms”. We hope this title better captures the essence of the manuscript’s take-home message.

Reviewer 1, point 10: *“I conclude by noting that the source of the result in Figure 3A was non-obvious to me. One can write out all the possible mathematical scenarios and this leads to clear expectations for the concentration of NPM1 in the light phase, the slopes of tie lines, and the partitioning of PEG into / out of the droplet. It is based on this analysis that I furnish the predictions laid out above. Data speak for themselves. The key point is that the support for the model being articulated here is incomplete absent the measurement of [NPM1] in the light phase and investigating the coupling between increase / decrease / stasis of this concentration vis-a-vis the data shown in Figure 3A. Either the authors are absolutely right and crowders influence the strengths of homotypic interactions via depletion effects and nothing else (i.e., the crowders are inert) or there is selective partitioning of PEG within the droplets and this contributes to the crosslinking as well as modulation of the affinities between acidic and basic tracts.”*

Authors: We thank the reviewer for pointing out the importance of reporting the protein concentrations in the light phases, in addition to those originally reported for the dense phases. The quantification of the NPM1 concentration in the light phase is now included in Fig. 3A, and the corresponding methodology is described in the *Materials and Methods* section (page 18).

Responses to Reviewer #2

Reviewer #2, point 1: *“Ferrolino et al. studies different aspects of co-phase separation of two nucleolar proteins: NPM-1 and SURF6, including the effect of molecular crowding, measurement of material properties, the influence of SURF-6 on the concentration and dynamics of NPM-1 within phase-separated condensates. The*

authors convincingly show that phase separation dependent on homotypic NPM-1-NPM-1 interaction generates condensates that differ in composition and dynamics from condensates with both homotypic NPM-1-NPM-1 and heterotypic NPM-1-SURF-6 interactions.

Co-phase separation involving heterotypic interactions between two proteins or between a protein and an RNA have been well studied in other systems (Banani et. al. Cell 2016, Zhang et. al. Mol Cell 2015 for example). This study with the nucleolar proteins does not offer significantly novel insights in terms of broadly thinking about co-phase separation of two macromolecules.

The authors propose that some aspects of co-phase separation between NPM-1 and SURF-6 can control the directionality of ribosome biosynthesis within the nucleolus. This claim is too preliminary and needs to be substantiated with significant in vivo and in vitro experiments.”

Authors: We thank the reviewer for their comments. We recognize that complex coacervation between two proteins and a protein and RNA, and compositional control within multi-component droplets, have been previously studied. However, we believe that our manuscripts offers novel insights in two areas: (1) it provides the first mechanistic insights into the role of SURF6 in maintaining and modulating the liquid-like structure of the nucleolus, thereby explaining previous data showing that SURF6 knock-down causes defects in ribosome biogenesis (PMID: 16855206), and (2) it builds support for a model wherein compositional inhomogeneities within the nucleolar microenvironment, which our data suggest are possible due to NPM1’s adaptability, may support the directionality of ribosome assembly from the inside to outside of the nucleolus.

As the reviewer correctly stated, the idea that coacervation between NPM1 and SURF6 may control the directionality of ribosome biogenesis is proposed as a hypothesis and we clearly articulate this in the last paragraph of the manuscript, which closes with the following sentence (on pages14-15):

“Future studies will be required, however, to test our hypotheses regarding how the different types of competitive scaffolds, involving numerous proteins and nucleic acids, influence the molecular rearrangements within the nucleolus that accompany vectorial ribosome biogenesis.”

We acknowledge that the last paragraph is speculative but feel that we clearly identify speculative statements as such. How phase separation contributes to ribosome subunit assembly in the nucleolus is a major open question in nucleolar biology and there are very few mechanistic studies in the literature that speak to this question. The new data in this manuscript have stimulated ideas regarding this question and we feel it is appropriate, at the very end of the paper, to share these ideas with readers. We do not claim that we have answers but rather offer possibilities as to how the physical effects of SURF6 on LLPS by NPM1 may contribute to creating gradients within the nucleolus that

may, in some way, contribute to, or even arise as a consequence of, vectorial ribosomal particle assemble. We respectfully request to leave the final paragraph essentially unchanged. We did modify it, however, to include a new sentence regarding our published model for NPM1's role in vectorial ribosome assembly, as follows (page 14).

“A model for how NPM1's multiple mechanisms of LLPS may mediate vectorial ribosomal subunit assembly was previously discussed⁵.”

We also changed the final sentence of the Abstract to reflect the speculative nature of these concepts:

(Page 2) “We propose a mechanism wherein NPM1-dependent nucleolar scaffolds are modulated by non-ribosomal proteins through active rearrangements of interaction networks that can possibly contribute to the directionality of ribosomal biogenesis within the liquid-like nucleolus.”

Reviewer #2, point 2: “1) How crowding affects dynamics in condensates needs to be clarified. A) The authors claim that PEG crowding works through volume exclusion effects. Does PEG concentrate within the condensates? How does phase separation respond to crowding with PEG of different sizes? B) The authors claim that PEG crowding increases NPM-1-NPM-1 interactions within homotypic droplets and stabilization of these interactions over time leads to maturation of condensates. Why is maturation faster at lower concentration of PEG (see Figure 3C)? Does PEG play a more significant role rather than being a passive crowding agent?”

Authors: We thank the reviewer for raising the issue of PEG playing a potential role as a scaffold within NPM1 droplets. In a series of control experiments, we demonstrated that PEG nearly equally partitions between NPM1 dense and light phases (Supplementary Fig. 5 and Table S1), and that PEG crowders with chain lengths equal to and larger than 4K Da induce NPM1 homotypic LLPS (Supplementary Fig. 6). Reviewer #1 also raised these issues and we ask Reviewer #2 to please read our detailed reply above (**Reviewer #1, point 4**).

With respect to the kinetics of droplet aging under different PEG percentages, the reviewer is correct; the 15% PEG droplets undergo gelation in a shorter time frame than those made with 5% PEG. Please see our reply to **Reviewer 1, point 6**, above, which addresses this issue.

Reviewer #2, point 3: “2) The conclusion that condensates containing both NPM-1 and SURF-6 have an NPM-1 rich shell (boundary) is not convincing. In the condensate presented in Figure 5G, while the right-side boundary looks green, the left-side boundary looks yellow. In the quantification presented in Figure 5J, NPM-1 appears to be only ~50% more enriched at the interface compared to SURF-6. Have the authors considered the possibility that the interface contains about the same number of molecules of NPM-1 and SURF-6, while the observed 50% difference is due to the use of different fluorophores to label NPM-1 and SURF-6. The figure legends for 5I-J should

describe more completely what the reaction mix contains.”

Authors: In order to strengthen the point that the observed ring is not the result of an optical aberration, we now include in the manuscript new results involving quantification of the normalized fluorescence intensity ratios between NPM1 and S6N in the Z-plane, in comparison with the same analysis performed on standard, fluorescent beads (Supplementary Fig. 15). We also included the following clarifications in the text, which more fully explain the radial image analysis algorithm:

Page 11: “We analyzed the spatial heterogeneity in composition within NPM1-S6N droplets using a custom image analysis algorithm termed Sauron. Sauron radially normalizes the fluorescence intensities for both NPM1 and S6N from the droplet center to the boundary (Fig. 5H), and then averages these values (Fig. 5I). To determine compositional variations from the center to the boundary of droplets, we calculated the ratio of normalized radial average fluorescence intensity values for NPM1 and S6N (Figure 5J, black trace). We found that the boundary of the droplet was apparently enriched with NPM1 compared to the interior (Fig. 5J). This inhomogeneity in compositions was also evident in reconstructed Z-stack images of NPM1-S6N droplets (Supplementary Fig. 15A-B). To validate that this was not an artifact from chromatic aberrations, we imaged homogeneously fluorescent beads of comparable size using identical acquisition parameters. We show that a notable heterogeneity exists within the droplet compared to the standard beads where the NPM1:S6N normalized fluorescence intensity ratio increases with increasing distance from the slide surface (Supplementary Fig. 15C). Furthermore, we consistently observed this heterogeneity on both hydrophobic and hydrophilic surfaces, suggesting that it was not due to preferential interaction with the slide surface (Supplementary Fig. 15B, D).”

Materials and Methods (page 20): “We have established that the heterogeneity in the fluorescence distributions for NPM1-A488 and S6N-A647 were not artifacts from chromatic aberrations by imaging standard homogeneously fluorescent 4- μ m beads (Thermo Fisher Scientific, Waltham, MA) using the same imaging parameters as used for the droplets.”

Reviewer #2, point 4: “3) Could the authors repeat some of their experiments with a mutant of SURF-6 or NPM-1 that weakens/strengthens the interaction between the two proteins? This will help strengthen the claim that the observed differences are indeed due to binding between SURF-6 and NPM-1.”

Authors: In one of our recent publications (Mitrea, *et al.*, *Nature Communications*, 2018) we showed that mutations in acidic tract 3 disrupt interactions with S6N to the extent that no phase separation is observed, and mutations in basic tract 2 disrupt homotypic interactions, increasing the saturation concentration for NPM1:S6N phase separation. We refer the reviewer to this published report for this information.

Reviewer #2, point 5: “1) The word “cross-linking” is often used to describe “interactions” in condensates. Unless there is evidence that crosslinking occurs, the

word should be revised.”

Authors: We acknowledge reviewers point regarding the potential confusion that the term “crosslinking”, which we used to indicate that S6N non-covalently binds two different NPM1 pentamers, and have replaced the term “crosslinking” with “interaction” throughout the manuscript. We thank the reviewer for noting this.

Reviewer #2, point 6: “2) The text often reads as if S6N is a scaffold while it is actually a client. The scaffold NPM-1 comes in two forms – in complex with itself or with S6N. Revision could be considered.”

Authors: We respectfully disagree and feel that the term scaffold applies to S6N for the following reasons. (1) Fig. 5C & D in this manuscript shows S6N displacing NPM1 from the homotypic scaffold, and assuming a scaffolding role itself in so doing, and (2) Fig. 3 in (Mitrea; *et al*, *Nature Communications*, 2018) shows a truncation mutant of NPM1 incapable of homotypic phase separation (NPM1^{N188}) undergoing heterotypic LLPS with S6N; the resulting droplets have a heightened partition coefficient for S6N and a decreased partition coefficient for NPM1^{N188}, compared to wild-type NPM1:S6N droplets. These latter observations also indicate a scaffolding role for S6N.

Reviewer #2, point 7: “3) Figures 2, 4: The authors should clarify in the figure itself or figure-legend about the configuration of the FRAP experiment – i.e. a small internal area bleached to probe for internal rearrangement of macromolecules vs. bleaching a whole condensate to additionally monitor exchange of macromolecules between condensate and surroundings.”

Authors: We now include the following clarification in the figure captions for Fig. 2D, 3C, 4A: “ROI = 1 μm circular area in the center of the droplet.”

Reviewer #2, point 8: “4) Are there any estimates available regarding the viscosity of nucleolus granular component in cells? If yes, how does it compare with the values measured here by the authors?”

Authors: The estimated viscosity value for the granular component (Feric, *et al.*, *Cell*, 2016) is ~12 mPa·s (determined from measured values of the inverse capillary velocity and surface tension). While the viscosity of the granular component and *in vitro* droplets are within the same order of magnitude, it is reasonable that the viscosity of the granular component is larger than that of *in vitro* droplets, given the greater compositional complexity of the natural organelle.

References

- 1 Wei, M. T. *et al.* Phase behaviour of disordered proteins underlying low density and high permeability of liquid organelles. *Nat Chem* **9**, 1118-1125, doi:10.1038/nchem.2803 (2017).

- 2 Soranno, A. *et al.* Single-molecule spectroscopy reveals polymer effects of disordered proteins in crowded environments. *Proc Natl Acad Sci U S A* **111**, 4874-4879, doi:10.1073/pnas.1322611111 (2014).
- 3 Patel, A. *et al.* A Liquid-to-Solid Phase Transition of the ALS Protein FUS Accelerated by Disease Mutation. *Cell* **162**, 1066-1077, doi:10.1016/j.cell.2015.07.047 (2015).
- 4 Molliex, A. *et al.* Phase separation by low complexity domains promotes stress granule assembly and drives pathological fibrillization. *Cell* **163**, 123-133, doi:10.1016/j.cell.2015.09.015 (2015).
- 5 Mitrea, D. M. *et al.* Self-interaction of NPM1 modulates multiple mechanisms of liquid-liquid phase separation. *Nat Commun* **9**, 842, doi:10.1038/s41467-018-03255-3 (2018).

REVIEWERS' COMMENTS:

Reviewer #1 (Remarks to the Author):

The manuscript is much improved. I am almost convinced that PEG is as inert as the authors propose it is, but a decisive verdict will have to wait some more thorny experiments and the development of a suitable theoretical framework for depletion-mediated interactions. This was my major concern, and I feel comfortable with the additional data and most of the verbiage. I would have crafted the narrative slightly differently, but that's a matter of choice, style, and bias. I believe the manuscript is ready to be published with a few minor tweaks.

1) I believe reviewer #2 is incorrect on the issue of physical crosslinks. These multivalent systems are instantiations of associative polymers and the valence of stickers does give rise to crosslinks, albeit of the non-covalent variety. In the interest of appeasing reviewer #2, the authors went away from a strong narrative to a weaker one. I ask that the term physical crosslinks be reinserted, at least once, so this work becomes a true exemplar of network fluids, which is what these systems really are.

2) The title is an improvement but I don't know that it makes things perfectly clear. I suggest additional tweaks.

3) On a semantic note, if we accept the definition of gelation as the formation of system spanning networks via physical crosslinks that cross a certain threshold, then the aging observed is not gelation. Rather, the correct term is hardening or aging.

Other than these three minor points, I have nothing more to add / request. This is a very important paper and it is likely to shape the thinking for what lies ahead.

Reviewer #2 (Remarks to the Author):

In the revised version of the manuscript, the authors have adequately addressed my concerns. I think researchers working on the nucleolus will find this work interesting. I have one reservation regarding presentation of data, but otherwise recommend publication.

Replace Figure 5I-J by a single panel (wider and therefore easy to see what is going on at the interface region). Instead of showing average data, plot individual values of Int NPM1/Int S6N similar to data currently shown in Fig 5H. This will allow readers to appreciate the distribution of Int NPM1/Int S6N values at different radial distances from the center of condensate.

POINT-BY-POINT RESPONSE TO REVIEWERS' COMMENTS

Authors: We thank both reviewers for their time and effort invested in reevaluating our manuscript, and we appreciate their positive feedback.

Although it doesn't change the manuscript narrative, we note that after correcting for the effect of viscosity on the fluorescence intensity of fluorescently-labeled PEG, the reported partitioning coefficient of PEG within NPM1-droplets has changed from slightly above 1 (1.1-1.3) to slightly below 1 (0.7-0.9). The text on page 6 now reads:

“Using TAMRA-labeled PEG-10K (PEG-TAMRA) as a probe, we measured partition coefficients of 0.7 ± 0.1 , 0.9 ± 0.1 and 0.9 ± 0.1 within homotypic NPM1 droplets formed in the presence of 5%, 15% and 30% PEG, respectively (Supplementary Fig. 5; Supplementary Table 2), indicating that PEG molecules do not favorably bind to NPM1 to form NPM1-PEG scaffolding interactions.”

Responses to Reviewer #1

Reviewer 1 – point 1: “I believe reviewer #2 is incorrect on the issue of physical crosslinks. These multivalent systems are instantiations of associative polymers and the valence of stickers does give rise to crosslinks, albeit of the non-covalent variety. In the interest of appeasing reviewer #2, the authors went away from a strong narrative to a weaker one. I ask that the term physical crosslinks be reinserted, at least once, so this work becomes a true exemplar of network fluids, which is what these systems really are.”

Authors: In order to acknowledge the perspectives of both reviewers, on page 6 of the manuscript we defined the inter-molecular interactions formed within the NPM1-S6N scaffold as non-covalent crosslinks. The text now reads as follows:

“For example, an increase of the NPM1-S6N droplet viscosity by 100-fold—much greater than that associated with 15% PEG (see Table 1)—would still allow for 15% NPM1 fluorescence recovery in the same experimental time frame (Supplementary Fig. 4); consequently, we propose that the reduced NPM1 mobility associated with crowding is due to increased inter-molecular interactions, which effectively mediate non-covalent crosslinking within the scaffold of the NPM1-S6N droplets.”

Reviewer 1 – point 2: “The title is an improvement but I don't know that it makes things perfectly clear. I suggest additional tweaks.”

Authors: We further revised the title to read as follows: “Compositional adaptability in NPM1-SURF6 scaffolding networks enabled by dynamic switching of phase separation mechanisms”

Reviewer 1 – point 3: “On a semantic note, if we accept the definition of gelation as the formation of system spanning networks via physical crosslinks that cross a certain threshold, then the aging observed is not gelation. Rather, the correct term is hardening or aging.”

Authors: We thank the reviewer for the explanatory note on terminology. We now replaced the term “gelation” with “aging” throughout the manuscript.

Responses to Reviewer #2

Reviewer 2: “Replace Figure 5I-J by a single panel (wider and therefore easy to see what is going on at the interface region). Instead of showing average data, plot individual values of Int NPM1/Int S6N similar to data currently shown in Fig 5H. This will allow readers to appreciate the distribution of Int NPM1/Int S6N values at different radial distances from the center of condensate.”

Authors: We thank the reviewer for their suggestion. Given the imperfections in the droplet radial symmetry, which are evident from Fig. 5G-H and SI Fig. 16, we feel that it is important to represent the fluorescence intensity distribution data as an average, with the associated standard deviation.